

# Aerosol water parameterization:
# long-term evaluation and importance

Swen Metzger[1,2,a], Mohamed Abdelkader[1,2,b], Benedikt Steil[2], and Klaus Klingmüller[2]

[1]Energy, Environment and Water Research Center, The Cyprus Institute, Nicosia, Cyprus
[2]Air Chemistry Department, Max Planck Institute for Chemistry, Mainz, Germany
[a]Now at: ResearchConcepts io GmbH, Freiburg im Breisgau, Germany
[b]Now at: King Abdullah University of Science and Technology, Saudi Arabia

**Correspondence:** Swen Metzger (swen.metzger@researchconcepts.io)

**Abstract.**

We scrutinize the importance of aerosol water for the aerosol optical depth (AOD) calculations by a long-term evaluation of the EQuilibrium Simplified Aerosol Model V4 for climate modeling, which was introduced by Metzger et al. (2016b). EQSAM4clim is based on a single solute coefficient approach that efficiently parameterizes hygroscopic growth, accounting

for aerosol water uptake from the deliquescence relative humidity up to supersaturation. EQSAM4clim extends the single solute coefficient approach to treat water uptake of multi-component mixtures. The gas-aerosol partitioning and the mixed solution water uptake can be solved analytically, preventing the need for iterations, which is computationally efficient. EQSAM4clim has been implemented in the global chemistry climate model EMAC and compared to ISORROPIA II (Fountoukis and Nenes, 2007) at climate time-scales. Our global modeling results show that (I) our EMAC results of the aerosol optical depth (AOD)

are comparable to independent results of Pozzer et al. (2015) for the period 2000-2010, (II) the results of various aerosol properties of EQSAM4clim and ISORROPIA II are similar and in agreement with AERONET and EMEP observations for the period 2000-2013, and (III) that the underlying assumptions on the aerosol water uptake limitations are important for derived AOD calculations. Sensitivity studies of different levels of chemical "aging" and associated water uptake show larger effects on AOD calculations for the year 2005 compared to the differences associated with the application of the two gas-liquid-solid

partitioning schemes. Altogether, our study reveals the importance of the aerosol water for climate applications.



# 1   Introduction

Providing realistic projections of climate change is one of the most difficult tasks for climate modelers, due to the many unknowns and large uncertainties that still exists (Intergovernmental Panel on Climate Change, 2014). For instance, the recent study by Klingmueller et al. (2016) suggests that the observed increase in aerosol optical depth (AOD) over large parts of the Middle East during the period 2001 to 2012 could to some extent prevail as a result of climate change. Even in absence of growing anthropogenic aerosol and aerosol precursor emissions, increasing temperature and decreasing relative humidity, as seen in the last decade, promote soil drying, which can lead to increased dust emissions and hence AOD. While this might be the case for arid regions all over the Earth, it is not an easy task for climate modelers to correctly quantify the effect due to the complexity of the underlying processes, as indicated by the studies of Abdelkader et al. (2015), Abdelkader et al. (2017).

To reduce uncertainties, the latter two studies applied the dust emissions scheme of Astitha et al. (2012) together with our chemical speciation of the emissions fluxes (see Section 2.4) in order to resolve a chemical aging of mineral dust particles (see Section 4.2). Furthermore, an interaction of the emission flux with meteorology (Klingmueller et al., 2018) and anthropogenic pollutants, together with a water mass conserving coupling of the aerosol hygroscopic growth into haze and clouds (Metzger and Lelieveld, 2007), is needed.

Proper hygroscopic growth calculations require thermodynamic models that can calculate at least the equilibrium partitioning of aerosols and their precursor gases from different natural sources in interaction with anthropogenic air pollution. To calculate the gas-liquid-solid phase partitioning, a variety of thermodynamic equilibrium models have been therefore developed (Metzger et al. (2016b) and references therein). For instance, MARS (Saxena et al., 1986) is widely used in regional modeling as the thermodynamic core of MADE/SORGAM (Ackermann et al. (1998), Schell et al. (2001)) through applications of the Weather Research and Forecasting model coupled to Chemistry (WRF-Chem, https://ruc.noaa.gov/wrf/wrf-chem/, Ahmadov and Kazil (2018)), the model of the European Monitoring and Evaluation Programme (EMEP, http://www.emep.int/, Simpson et al. (2012)), and the European Air Pollution Dispersion model system (EURAD, http://www.eurad.uni-koeln.de/). On the other side, for climate modeling mainly ISORROPIA (Nenes et al. (1998); Fountoukis and Nenes (2007)) and EQSAM (Metzger et al. (2002b), Metzger et al. (2006)) are widely used because of their computationally efficiency. Both codes (among others) were recently used for the investigation of global particulate nitrate as part of the Aerosol Comparisons between Observations and Models (AeroCom) phase III experiment (Bian et al., 2017). Besides this AeroCom study, different EQSAM versions have been used for various other modeling studies, e.g., EQSAM1 (up to EQSAM_v03d): Metzger et al. (2002b), Metzger et al. (2002a), Dentener et al. (2002), Lauer et al. (2005), Tsigaridis et al. (2006), Myhre et al. (2006), Luo et al. (2007), Bauer et al. (2007a) and Bauer et al. (2007b); EQSAM2: Trebs et al. (2005) and Metzger et al. (2006); EQSAM3: Metzger and Lelieveld (2007) and Bruehl et al. (2012). An overview of widely used modeling systems that provide an option to use either EQSAM and/or ISORROPIA is given in Table 1.

To reduce computational costs, both EQSAM and ISORROPIA follow the MARS approach (Saxena et al. (1986), Binkowski and Shankar (1995)) to determine certain domains by the degree of sulfuric acid neutralization and then divide the relative humidity (RH) and composition space into subdomains to minimize the number of equations to be solved. But in contrast to





EQSAM, all other thermodynamic equilibrium models require an iterative procedure to solve the ionic composition, which adds significantly to computational costs.

To accurately parameterise the aerosol hygroscopic growth by also considering the Kelvin effect as described by Metzger et al. (2012), the EQSAM approach (Metzger et al., 2002b)) was recently extended by Metzger et al. (2016b). The new model

version, the EQuilibrium Simplified Aerosol Model V4 for climate modeling, enables aerosol water uptake calculations of concentrated nanometer-sized particles up to dilute solutions, i.e. from the compounds relative humidity of deliquescence (RHD) up to supersaturation (Köhler theory). EQSAM4clim extends the single solute coefficient approach of Metzger et al. (2012) to multi-component mixtures, including semi-volatile ammonium compounds and major crustal elements. The advantage of EQSAM4clim is that the entire gas-liquid-solid aerosol phase partitioning and water uptake including major mineral cations

(Sec. 2.3), can now be solved analytically without iterations, which potentially significantly speeds-up computations on climate time-scales (Appendix B). Since the thermodynamics of the few widely used equilibrium models such as MARS are limited either to the ammonium-sulfate-nitrate-water system, or only include sodium and chloride but no crustal compounds such as calcium, magnesium and potassium, EQSAM4clim has been evaluated with its introduction against ISORROPIA II at various levels of complexity. It was shown by Metzger et al. (2016b) that the results of EQSAM4clim and ISORROPIA II are similar

for reference box-model calculations, textbook examples and 3D applications on time-scales of individual years.

To scrutinize the importance of aerosol water for climate applications, we evaluate the AOD calculations of EQSAM4clim and ISORROPIA II on climate time-scales. For this we extend the model evaluation of (Metzger et al., 2016b) by using the comprehensive chemistry-climate and Earth System model EMAC in a similar setup as applied in our studies on (I) the dust–air pollution dynamics over the eastern Mediterranean (Abdelkader et al., 2015), (II) the sensitivity of transatlantic dust transport

to chemical aging and related atmospheric processes (Abdelkader et al., 2017), and (III) the comparison of the Metop PMAp2 AOD products using model data (EUMETSAT ITT 15/210839, Final Report, Metzger et al. (2016a)). These studies employ a highly complex chemistry setup, particularly with respect to the gas-and aqueous phase chemistry and the associated chemical aging of primary aerosols. Since all three studies revealed the importance of chemical aging of primary dust particles for the calculation of the AOD, due to the regionally amplifcation by the aerosol water uptake, its important to evaluate the aerosol

water parameterizion also on climate time-scales. Our EMAC model setup is described in Section 2 and evaluated in Sec. 3 for three periods, 2005, 2000-2010 and 2000-2013 and different model setups that are scrutinized in Sec. 4. Additional results are presented in the Supplement. We conclude with Section 5.



## 2 Model description

### 2.1 Atmospheric Chemistry-Climate Model EMAC

We use the atmospheric chemistry-climate model EMAC following Abdelkader et al. (2015). EMAC comprises a numerical chemistry and climate simulation system that includes sub-models describing tropospheric and middle atmosphere processes and their interaction with oceans, land and human influences (Joeckel et al. (2005), Joeckel et al. (2006a), Joeckel et al. (2006b), Joeckel et al. (2008), Joeckel et al. (2010), Joeckel et al. (2016)). The core atmospheric model, i.e., the 5th generation European Centre Hamburg general circulation model (ECHAM5, Röckner et al., 2006), is applied with a spherical truncation of T42 and T106 (Gaussian grid of $\approx 2.8$ x $2.8^o$ and $\approx 1.1$ x $1.1^o$ in latitude and longitude) and 31 vertical hybrid pressure levels up to 10 hPa. Our model setup comprises the sub-models: AEROPT, AIRSEA, CLOUD, CLOUDOPT, CONVECT, CVTRANS, DDEP, GMXE, GWAVE, H2O, JVAL, LNOX, MECCA, OFFEMIS, ONEMIS, ORBIT, RAD, SCAV, SEDI, SURFACE, TNUDGE, TROPOP (http://www.messy-interface.org/).

Dry deposition (DDEP) and sedimentation (SEDI) are described by Kerkweg et al. (2006a) and are based on the big leaf approach Ganzeveld et al. (2006). Dry deposition velocities depend on physical and chemical properties of the surface cover. Wet deposition (SCAV) is described by Tost et al. (2006a), while its impact on atmospheric composition is detailed by Tost et al. (2006b) and Tost et al. (2007). The offline (OFFEMIS) and online (ONEMIS) emission calculations, including tracer nudging (TNUDGE), are described by Kerkweg et al. (2006b)). The oceanic DMS emissions, water isoprene concentration and methanol ($CH_3OH$) water deposition are calculated online with the sea-air exchange submodel (AIRSEA), the latter based on undersaturation of the oceanic surface water (Pozzer et al., 2006). The atmospheric chemistry is calculated with the chemistry submodel (MECCA), which was introduced with Sander et al. (2005).

Our chemical mechanism for the troposphere is similar to the one used in poz – initially described in Joeckel et al. (2006a) (see electronic supplement), although we use a reduced chemistry setup, which consists only of 40 (instead 104) gas phase species and of only 80 (instead 245) chemical reactions. $O_3$ related chemistry of the troposphere is well included, but we exclude decomposition of non-methane-hydrocarbons (NMHCs) (von Kuhlmann et al., 2003). The other sub-models used in this study are CONVECT (Tost et al., 2006b), LNOX (Tost et al., 2007), as well as CLOUD, CVTRANS, JVAL, TROPOP, H2O, ORBIT, and RAD (Joeckel et al., 2006a). The aerosol radiative properties (AEROPT) (poz, Klingmueller et al. (2014)) are based on the scheme by Lauer et al. (2007). AEROPT takes the width and mean radii of the lognormal modes into account and considers the composition to obtain the extinction coefficients ($\sigma_{sw,lw}$), single scattering albedo ($\omega_{sw,lw}$) and asymmetry factors ($\gamma_{sw,lw}$) for the shortwave (sw) and longwave (lw) radiation. The radiative forcing is fully coupled in our EMAC version with the aerosol water of primary and secondary aerosols, whereby the emission fluxes of primary particles is calculated online in feedback with the EMAC model meteorology (Sec. 2.4).

To represent the actual day-to-day meteorology in the tropospherehe, the model dynamics are weakly nudged (Jeuken et al. (1996), Joeckel et al. (2006a), Lelieveld et al. (2007)) towards the analysis data of the European Centre for Medium-Range Weather Forecasts (ECMWF) operational model data (up to 100 hPa). This allows a direct comparison of the model chemistry with ground station and satellite observations (Sec. 3). Our model emissions are kindly provided by the anthropogenic emission





inventory EDGAR-Climate Change and Impact Research (CIRCE) (Doering et al. (2009a), Doering et al. (2009b), Doering et al. (2009c)) on a high spatial (0.1 by 0.1$^o$) and moderate temporal (monthly) resolution – see e.g., poz and Pozzer et al. (2017) for details.

## 2.2 Aerosol Microphysics

Aerosol microphysics and the underlying gas-liquid-solid aerosol partitioning is calculated with the Global Modal-aerosol eXtension (GMXe) module, which was described by Pringle et al. (2010a) and Pringle et al. (2010b) but originally developed as part of Metzger and Lelieveld (2007). With GMXe we resolve the aerosol size distribution in seven, i.e., four soluble (nucleation, aitken, accumulation and coarse) and three insoluble (aitken, accumulation and coarse) log-normal modes. Primary particles are emitted in the insoluble modes (aitken, accumulation, coarse) and only transferred upon a chemical aging and

transport to the respective soluble modes (aitken, accumulation, coarse). Our description of "aging" depends on the amount of available condensable compounds that are the outcome of various emission processes (OFFEMIS, ONEMIS) and chemistry calculations (GMXe, MECCA, SCAV). For the chemical aging we follow our approach introduced with Abdelkader et al. (2015), which is scrutinzed in Section 4.2. The condensation dynamics are calculated within GMXe such that coagulation and hygroscopic growth can alter the aerosol the size-distributions. Small particles are efficiently transferred to larger sizes,

whereby hygroscopic growth of individual aerosol compounds is calculated from aerosol thermodynamics (Sec. 2.3) based on a chemical speciation of the aerosol emission fluxes (Sec. 2.4). Water uptake of bulk particles (OC, BC, SS, DU), which can be optionally considered, is only treated for aged particles in the soluble modes (Sec. 2.5). Additionally, our EMAC version allows to consider the aerosol hysteresis effect (Sec. 2.6). To avoid an overlap with cloud formation (especially optical thin clouds) the availability of water vapor is dynamically determined within GMXe. This limits the aerosol hygroscopic growth calculation

by either ISORROPIA II or EQSAM4clim, described in Sec. 2.3. Through this specific dynamical coupling, our overall water uptake process depends on meteorology and strongly alters with altitude, independently of the aerosol composition.

## 2.3 Aerosol Thermodynamics

Aerosol thermodynamics is represented by EQSAM4clim (Metzger et al., 2016b) and ISORROPIA II (Fountoukis and Nenes, 2007). Both gas-aerosol partitioning routines calculate the gas-liquid-solid partitioning and aerosol hygroscopic growth. They

are embedded in GMXe in exactly the same way, so that a direct comparison of the EMAC modeling results can be made. Deviations can be fully explained by differences in the EQSAM4clim and ISORROPIA II composition calculation approach. Both, EQSAM4clim and ISORROPIA II offer a computationally efficient treatment of the multi-component and multi-phase gas-liquid-solid aerosol partitioning at regional and global scales, by dividing the relative humidity (RH) and composition space into subdomains that minimize the number of equations to be solved. However, the EQSAM4clim framework is based

on a single solute specific coefficient ($v_i$), which was introduced by Metzger et al. (2012) to efficiently parameterise the water uptake of concentrated nanometer-sized particles up to dilute solutions. In contrast to ISORROPIA II, EQSAM4clim covers the mixed solution hygroscopic growth considering the Kelvin effect, i.e. water uptake from the compound's relative humidity of deliquescence (RHD) up to supersaturation (Köhler theory). It was shown by Metzger et al. (2016b) that the $\nu_i$-approach allows




to analytically solve the gas-liquid-solid partitioning and the mixed solution water uptake by eliminating the need for numerical solutions, which can significantly speed-up our EMAC computations (Appendix B). For a consistent model inter-comparison, we limit in this study the gas-aerosol partitioning and associated hygroscopic growth of our EMAC simulations to the inorganic compounds considered by ISORROPIA II. That is, we consider the gas-liquid-solid aerosol partitioning and water uptake of

the precursor gases water vapor ($H_2O$), sulfuric acid ($H_2SO_4$), nitric acid ($HNO_3$), hydrochloric acid (HCl), ammonia ($NH_3$), together with the major cations sodium ($Na^+$), potassium ($K^+$), calcium ($Ca^{2+}$), magnesium ($Mg^{2+}$), ammonium ($NH_4^+$), and the major anions sulfate ($SO_4^{2-}$), bisulfate ($HSO_4^-$), nitrate ($NO_3^-$), chloride ($Cl^-$). To enable the full complexity of the phase partitioning with EQSAM4clim and ISORROPIA II, we extend the default EMAC setup through ions assigned to the emission fluxes of primary aerosol particles.

## 2.4 Chemical speciation of aerosol emission fluxes

We extend our EMAC setup to include a basic chemical speciation of the natural aerosol emission fluxes in terms of certain cations and/or anions. Usually, climate models treat only bulk tracers such as sea salt (SS), dust (DU), organic carbon (OC) / black carbon (BC). Instead, we assign ions to the bulk emission fluxes of primary aerosols by using the major cations $Na^+$, $K^+$, $Ca^{2+}$, $Mg^{2+}$ and anions $SO_4^{2-}$, $Cl^-$. Our concept of chemical speciation was originally developed as part of GMXe by

Metzger and Lelieveld (2007) to extend the aerosol water uptake calculations to the so far chemically unresolved bulk aerosol mass. Thus, for bio-mass burning OC and BC aerosols, we consider the potassium cation ($K^+$) as a key reagent (proxy) for the water uptake thermodynamics (Sec. 2.3). For DU, we respectively consider as a chemical aging proxy the calcium cation ($Ca^{2+}$), while we resolve the sea salt emission fluxes in terms of the sea water composition, considering the major cations $Na^+$, $K^+$, $Ca^{2+}$, $Mg^{2+}$ and anions $Cl^-$ and $SO_4^{2-}$. Our emission fluxes of primary sea salt and dust particles are calculated online, in

feedback with the EMAC meteorology and radiation computations. Sea salt is emitted in two soluble modes (accumulation and coarse) based on the flux parameterization of Monahan et al. (1986), while mineral dust particles are emitted in two insoluble modes (accumulation and coarse), following Astitha et al. (2012). The required parameters for OC/BC, SS and DU used in our sensitivity study (Sec. 4) to scrutinze the bulk water uptake are given in Table 2 and described in Sec. 2.5.

### 2.5 Chemical aging and water uptake of bulk aerosols

Our chemical speciation of the primary aerosol emission fluxes is coupled to a chemical aging of bulk species through which salt compounds and associated water can be formed. The chemical aging process is hereby based on explizit neutralization reactions of ions (cations, or anions), which are assigned to the emission fluxes (e.g., $K^+$, $Ca^{2+}$, see Sec. 2.4). Through the reactions of these cations (anions) with aerosol precursor gases, i.e., major oxidation products of natural and anthropogenic air pollution (here $H_2SO_4$, $HNO_3$, HCl, $NH_3$, and $H_2O$), various neutralization (salt) compounds can be formed, e.g., potassium

sulfate ($K_2SO_4$), potassium bisulfate ($KHSO_4$), potassium nitrate ($KNO_3$), potassium chloride (KCl), calcium sulfate ($CaSO_4$), calcium nitrate ($Ca(NO_3)_2$), calcium chloride ($CaCl_2$) and so on for ammonium, sodium and magnesium, see Table 1 of Metzger et al. (2016b). The salts can cause an uptake of water vapor ($H_2O$) at different ambient humidities, with $CaCl_2$ at RHs as low as 28%. All salt solutions are subject to the RH and T–dependent gas-liquid-solid partitioning as described in Sec. 2.3



and 2.6. For H₂O and each cation and anion, a chemical tracer is assigned such that they undergo all aerosol microphysics and thermodynamic processes for their respective GMXe aerosol mode(s) (Sec. 2.2). Through this tracer coupling, each salt compound can alter the subsequent AOD calculations in our EMAC version, most noticeably through an associated aerosol water uptake.

To calculate the bulk water uptake, we use the EQSAM4clim parameterizations (introduced by Metzger et al. (2012)) and solve a bulk solute molality using Eq. A3 of Metzger et al. (2016b). For the sake of simplicity, we neglect the Kelvin-term ($K_e = 1$, $A = 1$, $B = 0$) and further assume that the water uptake of the bulk compounds can be described by a mean value, for which we can use our single coefficient $\nu_i$. We further assume a single chemical reagent to be representative for the bulk water uptake due to chemical aging of the bulk aerosol mass, but we only calculate bulk water uptake if the RH exceeds a certain

threshold. This "aging" proxy is given in Table 2 together with the required parameters for our "aging" setup used in Sec. 4.2. For instance, for the "50% aging" case of bulk sea salt mass (SS), we assume $50[\%]$ of the mass to be subject to water uptake if the RH exceeds a threshold of $50\%$. And for this case we assume NaCl as the proxy with $\nu_i = 1.358$ (Table 1 of Metzger et al. (2016b)). Accordingly, we assume for dust (DU) that $75\%$ of the mass is subject to water uptake if the RH exceeds the threshold of $28\%$, due to a pre-dominant coating by CaCl₂ (with $\nu_i = 2.025$).

To distinguish between our EMAC setup that considers the water uptake of normally chemically unresolved particles (SS, DU, OC, BC), we use in our study the label "$aging$", refering to a chemical "aging" that is used in Sec. 4.2. In contrast, our EMAC setup that omits the chemical "aging" and associated water uptake of bulk aerosols is labeled "$no\ aging$" (Sec. 4.1). Independent of this "aging" label, all our EMAC simulations consider a comprehensive treatment of the chemical aging of the non-bulk aerosol emission fluxes, which is part of our GMXe aerosol dynamical treatment Sec. 2.2. The chemical aging

includes the dynamically limited condensation of aerosol precursor gases on primary aerosol particles. Our primary aerosol particles are emitted in the insoluble modes and, depending on the coating level (i.e., the amount of gases condensed on the insoluble particles), they are transferred to the soluble modes. But only the chemically identified compounds of the soluble modes (aitken, accumulation and coarse mode) are subject to the water uptake calculations by either EQSAM4clim or ISOR-ROPIA II by our "$no\ aging$" set-up. Since the inorganic aerosol composition usually explains only a fraction of the emission

fluxes, and since the coating process may involve complicated and largely unknown chemical reactions which alter (age) the aerosol surfaces, we consider for our sensitivity study in Sec. 4 the water uptake of the bulk aerosol mass (as described above). Normally, the bulk aerosol mass would be otherwise considered as dry only. And it was shown by our recent studies by Abdelkader et al. (2015), Metzger et al. (2016b), Metzger et al. (2016a) and Abdelkader et al. (2017) that the results of our EMAC "$aging$" set-up agree better with various ground station observations and satellite measurements.

**2.6    Aerosol water mass – hysteresis effect**

Our EMAC version further allows to consider the so-called hysteresis effect. That is, we can obtain the aerosol water mass for two cases, i.e, (1) dry case, when RH increases and exceeds the compound's RHD, or mixed solution RHD (Sec. 2.6 of Metzger et al. (2016b)), and (2) wet case, when the RH decreases until crystallization (efflorescence) point of the dissolved compound(s) is reached. Below these thresholds no aerosol water is calculated. The hysteresis effect can become regionally




important, since many inorganic salt compounds, which take up water at a given RHD-threshold, do not crystallize at the same threshold. The efflorescence thresholds are often observed to be much lower. Although the hysteresis effect might be less pronounced in ambient observations (simply because the aerosol composition usually changes over time due to transport and chemical reactions), the instantaneous effect on radiation can locally become important.

To consider the hysteresis effect in a climate model, we assume for the sake of simplicity (and because of missing measurements) no single compound efflorescence thresholds. Our criteria that determines a "wet case" or "dry case" instead depends on two factors: (i) an RH threshold and (ii) the existence of aerosol water mass from the previous time-step. In case aerosol water mass from the previous time-step is non-zero for the given time-step (and model grid box), and, if additionally the RH is above $40\%$ (fixed efflorescence value), we consider the upper hysteresis loop and only calculate the gas-liquid partitioning

with either EQSAM4clim or ISORROPIA II. Otherwise, we account for the full gas-liquid-solid partitioning (lower hysteresis loop). The water uptake is then based on deliquescence of single or mixed solutions as described in Metzger et al. (2016b). Note that the aerosol water mass is treated prognostically in our EMAC version Sec. 2.5. That is, we assign a model tracer for water vapor and for each aerosol mode to transport the different water masses. This allows to retrieve the required time information for a certain location on Earth, although we are only approximately able to distinguish between the upper or lower

hysteresis loop. Results of our EMAC setup that include the hysteresis effect are shown in Sec. 3 and 4.

## 3    Climate applications

To evaluate the hygroscopic growth calculations of EQSAM4clim and ISORROPIA II and to evaluate our EMAC version we focus on the AOD, since long-term observations are available for many regions of the Earth. The AOD, or extinction coefficient, is a measure of radiation scattering and absorption at different wavelengths and sensitive to the gas-liquid-solid partitioning and aerosol hygroscopic growth. We use ground-station observations from the AErosol RObotic NETwork (AERONET,

http://aeronet.gsfc.nasa.gov). Complementary, we use independent satellite observations from MODIS and MISR (both available from http://disc.sci.gsfc.nasa.gov/giovanni.) The comparison of model results against measurements includes the in-situ observations of the Clean Air Status and Trends NETwork (CASTNET, www.epa.gov/castnet/). CASTNET is a national air quality-monitoring network of the United States of America designed to provide data to assess trends in air quality, atmospheric deposition, and ecological effects due to changes in air pollutant emissions. For Europe, we use data of the European

Monitoring and Evaluation Programme (EMEP) (http://www.emep.int/). EMEP is a scientifically based and policy driven program under the Convention on Long-range Transboundary Air Pollution (CLRTAP) for international co-operation to solve transboundary air pollution problems (Tørseth et al., 2012). Our EMAC model evaluation is based on two model resolutions, i.e. T42 and T106 (Sec. 2.1). Most of our model output is based on 5-hourly averages, such that any full hour serves as averaging-interval center once within 5 days. An extension of our study to a more in-depth evaluation of the underlying aerosol

composition and neutralization levels will be presented separately.





### 3.1 EMAC AOD versus AERONET and Satellites

The EMAC hygroscopic growth calculations of EQSAM4clim and ISORROPIA II are first compared for the period 2000-2010 to independent AOD results of Pozzer et al. (2015) (PO2015). To give a compact but representative picture of our analysis, we focus on a selection of AERONET stations that represent different regions of the Earth. Figure 1 shows the

selected station locations, Fig. S1 (Supplement) the corresponding regions. Fig. 2 shows the results of the AOD comparison (from left to right, top to bottom): GSFC (North America), Sao Paulo (South America), Cape San Juan (Latin America), Capo Verde (West Africa), Canberra (Australia), Yekaterinburg (Siberia), Forth Crete (EMME), Dakar (West Africa), Yakutsk (Siberia), Amsterdam Island (Indian Ocean), Lampedusa (North Africa), Beijing (East Asia). Fig. 3 shows the corresponding Taylor diagrams (standard deviation and correlation coefficient) of the AOD comparison of EQSAM4clim, ISORROPIA II

and PO2015. The comparison includes different observations from independent satellite instruments, i.e, MODIS, MODIS-Aqua, MODIS-deep blue, MISR, SeaWIFS and ENVISAT, which are discussed in detail in our extended evaluation study. All satellite products and model results are compared against the AERONET observations for the period 2000-2010 (based on globally averaged seasonal means using a 5 hourly model output and accordingly averaged AERONET observations – details are given in Metzger et al. (2016a)). The corresponding scatter plots are shown in Figures S2–S4 of the Supplement and include

the statistics: root mean square error ($\mathrm{RMSE}$), correlation coefficient ($\mathrm{R}$), mean biased error ($\mathrm{MBE}$), standard deviation of the model results ($\sigma_{\mathrm{m}}$) and AERONET observations ($\sigma_{\mathrm{o}}$). The equations are given in Appendix A: Evaluation Metrics.

The comparison shows that the differences associated with the two partitioning schemes are smaller compared to the differences associated with the two different EMAC setups, i.e., our EMAC version with EQSAM4clim (orange circles) and ISORROPIA II (blue stars), and the independent PO2015 setup (pink crosses). But all AOD model results are relatively close

to the AERONET observations, despite the distinct different underlying approaches to obtain the mixed solution aerosol water uptake. The largest differences occur for regions which are dominated by mineral dust outbreaks, as indicated by the AERONET stations Capo Verde and Dakar (Fig. 2). The reason is that PO2015 uses prescribed dust emissions, while our setup calculates the dust emission fluxes online with the EMAC meteorology (Sec. 2.4). Although the same is true for the sea salt emissions, differences there are much less pronounced (see e.g., Amsterdam Island). The prescribed dust emissions basically

yield a mean dust concentration with a too low variability, which is reflected in a too low variability of the AOD results (see pink crosses for e.g. Dakar in Fig. 2). Contrary, our EMAC version results show too low minimum values for certain periods (e.g., for 2002–2008), but the magnitude of the seasonal cycle is much closer to the AERONET observations (black circles). On the other hand, the setup of PO2015 is based on the T106L31 resolution ($\approx$1.1 x 1.1$^o$), while our results are based on a T42L31 ($\approx$2.8x2.8$^o$) setup. Although, the coarser resolution somewhat affects the statistics of the analysis (see Supplement),

our results are also within the range of the satellite results when compared to the AERONET observations (Figure 3). Notably, spring and summer seasons are for our T42L31 setup better resolved than the winter months. Altogether, the results indicate that we may underestimate the chemical aging of bulk particles, which is therefore scrutinized in Sec. 4.2.





## 3.2 EQSAM4clim versus ISORROPIA II for 2000–2013

To further evaluate EQSAM4clim and ISORROPIA II, we compare the AOD and the total particulate matter (PM) that drives the model AOD with AERONET and EMEP observations for the period for 2000–2013. Figure 4 shows the AOD and PM time-series and the climatological year (14 years average) for the EMEP station Harwell and the AERONET Chilbolton
(United Kingdom, Fig. 1). The two sites lie within one model grid box and are chosen, since no other site provides long-term observations of both AOD and PM. Only Cabauw in the Netherlands, which is one of the few EMEP and AERONET super-sites, provides AOD and PM observations with some reasonable overlap as shown in Fig. 5. To complement the picture, the corresponding aerosol water ($H_2O$), which is associated with the total model PM, is also shown for EQSAM4clim and ISORROPIA II (but no observations are available).

Figure S5 in the Supplement (Sec. S1.2) shows the corresponding size-resolved PM, aerosol water, number concentration and wet radius for each aerosol mode: nucleation soluble (ns), aitken soluble (ks), accumulation soluble (as), coarse soluble (cs), aitken insoluble (ki), accumulation insoluble (ai), coarse insoluble (ci); ISORROPIA II (left column), EQSAM4clim (right column). The sum of the modes (for PM, $H_2O$) is identical to Figure 5.

Figure 6 further shows various PM time-series of EQSAM4clim and ISORROPIA II (top panels) in comparison with EMEP
stations, which provide long-term PM observations, i.e., Cabo de Creus, Hyytiala, Illmitz and Vreedepel. The station locations are shown in Fig. 1, the corresponding climatological year below each time-series (Fig. 6). The corresponding global aerosol PM and associated water ($H_2O$) distributions (14 years average) are shown in Figures 7 and 8: meridional means (left columns), zonal means (middle columns) surface distributions (right columns), ISORROPIA II (ISO2, top rows), EQSAM4clim (EQ4c, middle rows), together with the differences between both simulations (EQ4c–ISO2, bottom rows).

Both, AOD and PM model results nicely compare with various surface observation for the entire evaluation period (2000-2013). But, also the global surface and vertical distributions from both EMAC simulations are in close agreement for the aerosol PM and $H_2O$, which supports our previous finding (Sec. 3.1) that the difference between EQSAM4clim and ISORROPIA II are negligible on climate simulation time-scales.

Figure 9 further shows scatter plots of the model AOD versus AERONET observations for the period 2000-2013 and the year
2005. For each period, three different time averages are shown i.e., 5 hourly averages (full time resolution), monthly means and station means based on 537 AERONET stations all over the Earth (locations are shown in Fig. S1 of the Supplement). The statistics included in each panel summarizes the results and show that both EMAC simulations are comparable in terms of statistical key metrics, i.e. Root Mean Square Error (RMSE), Standard deviation ($\sigma$), Correlation Coefficient (R), Mean biased Error (MBE) (equations are given in Appendix A). Interestingly, the statistics of all time averages indicate that the results of
EQSAM4clim are slightly closer to the AERONET observations compared to ISORROPIA II. Note that Fig. 14 complements Fig. 9 with the results for 2005 by our EMAC "aging" set-up discussed in Sec. 4.2.



### 3.3 EQSAM4clim versus ISORROPIA II for 2005

In order to scrutinize this result, we zoom into a single location and compare the EMAC AOD of EQSAM4clim and ISOR-ROPIA II for the AERONET observations at Capo Verde for both, 5 hourly and monthly averages (Figure 10). Capo Verde is one of the more difficult stations because of the frequent Sahara dust outflows (Abdelkader et al., 2017). In our setup the

dust outflow is associated with elevated calcium loadings, which can cause differences in the subsequent sulfate/bisulfate neutralization (Sec. 4). Despite the slight underestimation of the AOD observations by both model simulations, the results of EQSAM4clim and ISORROPIA II are very close throughout the year. Even the distinct AOD peaks im May, which can be attributed to Saharan dust outbreaks, are well resolved at the 5-hourly output frequency, although the comparison based on monthly averages seems to be less impressive. Nevertheless, the absolute comparison is overall very good for a chemistry–

climate model.

To evaluate the aerosol composition which drives the hygroscopic growth, we further compare our aerosol ammonium ($NH_4^+$) results against EMEP observations at the measurement site Vredepeel. $NH_4^+$ is the waekest cation considered in our simulations and driven out of the aerosol phase by all non-volatile cations, because of its semi-volatility. It is one of the most difficult aerosol species to model, if the mineral cations $Na^+$, $K^+$, $Mg^{2+}$ and $Ca^{2+}$ are considered, e.g., through a chemical

speciation of the aerosol emission fluxes (Sec. 2.4). $NH_4^+$ therefore shows for cation rich locations the largest sensitivity in our aerosol calculations (shown e.g. by the results of Sec. S1.3 in the Supplement). Only in case $NH_4^+$ is the only cation that neutralizes the anions $SO_4^{2-}$, $HSO_4^-$, $NO_3^-$, $Cl^-$, it is preferentially bound with sulfate for which the aerosol concentrations are usually in good agreement with observations. But, including mineral cations through a chemical speciation of emission fluxes, complicates the modeling enormously. Despite these challenges, our comparison with observations in Fig. 11 shows that the

total particulate ammonium, i.e., the sum of all liquid and solid $NH_4^+$ cations, compares well for different time averages for the year 2005. Differences between EQSAM4clim and ISORROPIA II are also rather small for the daily, monthly and even a 14 years monthly mean (climatological year).

To further evaluate our EMAC results on a global scale, Fig. 12 compares the annual mean AOD of ISORROPIA II (left panel) and EQSAM4clim (right panel) against AERONET observations (included as squares) for 2005 (upper and middle

row). The upper row represents our "*no aging*" case and excludes aging and hysteresis effects (Sec. 4.1), while the (middle row) represents our "*aging*" case and includes both effects (they are discussed further in Sec. 4.2). The lower row shows independent satellite observations from MODIS and MISR. Altogether, this comparison shows that the EMAC results based on EQSAM4clim and ISORROPIA II are also very similar on a global scale, and that the EMAC results labeled "aging" compare better with the satellite observations than the "no aging" results. This qualitative comparison indicates that the overall

assumption on the water uptake is important. But it also shows that the differences between the two different EMAC setups (comparing upper and middle row) are larger than the differences between the two distinct different gas-aerosol partitioning schemes (comparing left and right panels).



## 4 Sensitivity study (Year 2005)

To scrutinize the importance of the aerosol water calculations we compare our EMAC results in a sensitivity study that excludes (Sec. 4.1) and includes (Sec. 4.2) the aerosol water and bulk water uptake (Sec. 2.5) due to the chemical aging of primary particles (Sec. 2.4).

### 4.1 EMAC setup – without aging

Our EMAC setup without aging omits the water uptake of bulk aerosols (OC, BC, SS, DU) in contrast to the "aging" case (Sec. 4.2). For both setups we consider the chemical speciation of the emission fluxes (Sec. 2.4) to obtain chemically specified aerosol mass fractions in terms of cations and anions. But for the "no aging" case, we limit the water uptake to the neutralization products (ion pairs), which are calculated with the partitioning schemes (Sec. 2.3). Our reasoning for this limited setup is that the aerosol water mass of bulk species (Sec. 2.5), as well as the hysteresis effect (Sec. 2.6), can regionally reduce potential differences of the aerosol water mass calculations, if the total aerosol water mass is dominated by one of these effects. For both processes explicit RHD calculations and the associated uncertainties (Metzger et al., 2016b) are excluded. The "no aging" setup is therefore most sensitive to potential differences in the water uptake calculation approaches of EQSAM4clim and ISORROPIA II, though differences are rather small on a global scale as discussed in Sec. 3.3 (i.e., shown by the comparison in Fig. 12, upper panels).

We note that the relatively largest deviations occur in our "no aging" EMAC setup for stations that are subject to high dust loads, e.g., Dakar and Capo Verde (see Supplement). But the aerosol properties that are most important for climate modelling, i.e., the total (dry) PM and the associated aerosol water mass concentrations are mostly close to a one-by-one line for all simulations and all stations. Differences are mainly caused by differences in the bisulfate / sulfate partitioning of both schemes. In contrast to ISORROPIA II, EQSAM4clim does not treat the dissolution of weak acids ($HNO_3$, $HCl$) and bases ($NH_3$), which can cause differences in the sulfate neutralization levels and the subsequent water coating of mineral dust particles. Also the Kelvin effect is not considered in ISORROPIA II in contrast to EQSAM4clim, which can have an effect on the water uptake of Aitken mode but not coarser particles. Nevertheless, overall differences are small in terms of mass concentrations as shown by the extended analysis included in the Supplement.

Note that the Supplement (Sec. S1.3) shows both time series and scatter plots for 2005 for our "no aging" case which are based on all 537 AERONET stations of Fig. 1. The results include the PM (Fig. S6) and $H_2O$ (Fig. S7) concentrations $[\mu g/m^3(air)]$, as well as those of the lumped aerosols, i.e., sulfate ($SO_4{}^{2-}$), bisulfate ($HSO_4{}^-$), nitrate ($NO_3{}^-$), chloride ($Cl^-$), ammonium ($NH_4{}^+$), sodium ($Na^+$), potassium ($K^+$), magnesium ($Mg^{2+}$) and calcium ($Ca^{2+}$); shown in Figs. S8–S16. The corresponding scatter plots Fig. S17 – S20 show the annual means for three soluble (key) aerosol modes of GMXe (Sec. 2.2): coarse (top row), accumulation (middle row) and aitken (bottom row) and include the growth factor (GF, see Metzger et al. (2016b)). Each panel includes the statistics: Root Mean Square Error ($\mathrm{RMSE}$), Correlation Coefficient ($\mathrm{R}$), Mean biased Error ($\mathrm{MBE}$), Standard deviation of ISORROPIA II (x-STD) and EQSAM4clim (y-STD). Table 3 complements the time series and scatter plots with some additional statistics of key EMAC tracers.



## 4.2 EMAC setup – with aging

The EMAC setup labeled "aging" extends the "no aging" setup by the water mass calculation of bulk aerosol species (Sec. 2.5) and the hysteresis effect (Sec. 2.6) – note Table 4. Both can become regionally important. As noted in Sec. 3.3, our EMAC "aging" setup compares better with observations than the "no aging" case. This is especially true for regions over the open

oceans, intense bio-mass burnings or dust outbreaks, including the transatlantic dust transport as shown in Fig. 12. But despite the more complex "aging" setup, our EMAC version still somewhat underestimates the AOD observations. This finding is supported by the AERONET observations, which are included in Fig. 12 (squares with the same color scale). One reason could be that our default "aging" set-up only considers a partial "aging" of 50% of the bulk aerosol mass for the additional water uptake calculations.

To scrutinize the effect of "aging level" on the AOD comparison, we apply different levels of bulk "aging" according to Table 4. Figure 13) shows the results of four different EMAC simulations, i.e., case 1: "no aging" (blue stars), case 2: "no water" (orange circles), case 3: "50% aging" (pink crosses), case 4: "90% aging" (light blue squares). The upper two rows compare the model results of EQSAM4clim and ISORROPIA II based on case 4 for the AERONET observations at Lampedusa and Beijing for the year 2005. The first and third row show the 5 hourly means, while the 2nd and 4th row show the corresponding

monthly means. The lowest two rows present the key-results of our sensitivity study.

The comparison of the cases 1–4 shows that aerosol water calculations are essential. Excluding "aging" or aerosol water at all, our EMAC simulation largely underestimates the AOD (case 1–2), while considering the bulk water uptake ("aging" case 3–4) improves the AOD comparison. But, the improvement strongly depends on the AERONET location and the assumed level of "aging". For instance, our EMAC results based on a 90% "aging" level (case 4) can overestimate the AOD observations at

certain locations such as for Lampedusa, while at the same time the results compare best with other observations such at the AERONET site of Beijing. With a decreasing level of "aging", the AOD observations get more underestimated for Beijing, while improved for Lampedusa. This fact points to missing processes that cannot be resolved by applying constant aging parameters. To improve our results further, a more comprehensive aging parameterization is needed by e.g., an extension of the water uptake framework to organic compounds as considered by Metzger and Lelieveld (2007). The latter study included the

neutralization of major carboxylic acids for the neutralisation by the cations $Na^+$, $K^+$, $Ca^{2+}$, $Mg^{2+}$ to form salt compounds (formates, acetates, oxalates, citrates, see their Table 1), which can contribute to the overall aerosol water mass and hence regionally improve the AOD. Yet, such extensions are beyond the scope of this work. Here, we focus on a consistent model-inter comparison of EQSAM4clim and ISORROPIA II and the importance of aerosol water mass for the model evaluation in terms of AOD. Nevertheless, our EMAC results based on the higher "aging" level (case 4) improve the global scale comparison

of Fig. 9 (discussed in Sec. 3.2) as shown by Fig. 14.

## 4.3 Importance of Aerosol Water

The sensitivity of our AOD calculations with respect to the RH cut-off is analyzed next. Such a cut-off is required for all aerosol water mass calculations and applied to prevent overlap between aerosol hygroscopic growth and parameterized cloud





formation. Here we consider four different RH cut-off cases for which AOD results (2005, annual mean) of four EMAC simulations are shown in Figure 15 (from left to right, top to bottom): (UL) RH= 0[%], i.e., no aerosol water, (UR) RH= 97[%], (LL) RH= 98[%], and (LR) RH= 99.9[%]. The four simulations only differ by the assumption on the aerosol water uptake limitation, i.e., the upper RH value that is used to limit the water uptake calculation for both EQSAM4clim and ISORROPIA II.

While our first and last sensitivity simulations represent an extreme case (with unrealistic AOD results), the two simulations with RH= 97 and 98[%] cut-offs yield similar AOD results that are relatively close to many AERONET observations (colored squares). Noticeably, the AOD values significantly increase for the high RH= 99.9[%] case. Of course, any RH cut-off is arbitrary, if the aerosol water mass is not consistently linked with cloud formation. To avoid an inconsistent aerosol-cloud-radiation coupling, Metzger and Lelieveld (2007) proposed a mass conservative coupling approach to limit the aerosol water

mass by an approach that needs to be further scrutinzed too (presented elsewhere).

## 5   Conclusions

The importance of aerosol water for AOD calculations has been scrutinized by a long-term evaluation of EQSAM4clim and ISORROPIA II on climate time-scales using our EMAC model version as applied in Abdelkader et al. (2015), Metzger et al. (2016b) and Abdelkader et al. (2017). Generally, the results of both gas-liquid-solid partitioning schemes are in good agree-

ment despite differences in the bisulfate partitioning and mixed solution deliquescence humidity range, where the results of thermodynamic schemes are typically associated with deviations (Metzger et al., 2016b). However, these discrepancies are negligible for climate simulations, as the total aerosol water mass and AOD do not significantly differ. Furthermore, besides the relative importance of (a) the general model setup (EQSAM4clim or ISORROPIA II), (b) number and types of compounds considered for the aerosol water calculations (e.g., mineral cations), (c) water uptake by bulk species and chemical aging, (d)

hysteresis effect (efflorescence versus deliquescence), it appeared that (e) the aerosol water uptake limitations of both partitioning schemes is most determinant for AOD calculations. Overall, the comparison of our EMAC results with remote sensing AOD observations reveals the importance of the aerosol water calculations for climate applications.

*Acknowledgements.* This work was supported by the European Research Council under the European Union's Seventh Framework Programme (FP7/2007-2013)–ERC grant agreement no. 226144 through the C8-Project, and by the Energy oriented Centre of Excellence

(EoCoE), grant agreement number 676629, funded within the Horizon2020 framework of the European Union. EMAC simulations have been carried out on the supercomputer of the German Climate Research Center and on the Cy-Tera Cluster, operated by the Cyprus Institute (CyI) and co-funded by the European Regional Development Fund and the Republic of Cyprus through the Research Promotion Foundation (Project Cy-Tera NEA-$Y\Pi O\Delta OMH/\Sigma TPATH$/0308/31). We thank the measurement and model development teams for providing the observations, numerical models and the many useful codes that have been used in this study.



## Appendix A: Evaluation metrics

Root Mean Square Error:

$$RMSE = \sqrt{\frac{1}{N}\sum (X_m - X_o)^2} \tag{A1}$$

Standard deviation:

$$\sigma = \sqrt{\frac{1}{N}\sum_{i=1}^{N}(X_i - \bar{X})^2}, \qquad where \quad \bar{X} = \frac{1}{N}\sum_{i=1}^{N}X_i \tag{A2}$$

Correlation coefficient:

$$R = \frac{\sum_{i=1}^{N}(X_i^m - \bar{X^m})(X_i^o - \bar{X^o})}{\sqrt{\sum_{i=1}^{N}(X_i^m - \bar{X^m})^2 \sum_{i=1}^{N}(X_i^o - \bar{X^o})^2}} \tag{A3}$$

Mean biased Error (MBE):

$$MBE = \frac{1}{N}\sum (X_m - X_o) \tag{A4}$$

Index $m$ refers to EQSAM4clim (EQ4c) and $o$ to ISORROPIA II (ISO2).

## Appendix B: Computational Efficiency

Computational efficiency is a key-constraint on our model development. To scrutinize the model performance, we compare both gas-aerosol partitioning schemes (EQSAM4clim and ISORROPIA II) using the simulation period of 2005. Table 5 presents the computational burden (CPU times) for different EMAC simulations (T42L31). Experiments A, B, C and D correspond to a "*no aging*" EMAC setup (results of Exp A and B are shown in Sec. S1.3 of the Supplement). The four simulations only differ by the constraint on the gas/aerosol partitioning scheme, i.e., Exp A represents ISORROPIA II, Exp B and C represent two simulations of EQSAM4clim (identical setup, just quantifying numerical noise of the computing architecture), while for Exp D the call to the gas-liquid-solid partitioning scheme has been commented out, while all other GMXe processes remained unchanged (Sec. 2.2). Exp D therefore represents the minimum of CPU time that is required for our GMXe aerosol setup on the Cy-Tera super computer (https://cytera.cyi.ac.cy/). Two additional experiments, labeled Exp $A_0$ and $B_0$, represent sensitivity simulations of ISORROPIA II and EQSAM4clim, respectively. Both only omit anthropogenic emissions in our EMAC set up, while all other EMAC processes remained the same as for Exp A and B.

Table 5 reveals the real CPU-utilization. The comparison of the numbers shows: (i) a dependency of both partitioning schemes on the aerosol setup and composition (Exp A versus $A_0$ and B versus $B_0$), (ii) that the dependency of the additional computational costs for EQSAM4clim besides GMXe is small (Exp B and C versus D and $B_0$ versus D), while (iii) this is not that much the case for ISORROPIA II (Exp A versus D and $A_0$ versus D). Given the uncertainty in these numbers due to the different system loads (indicated by Exp B versus C), the additional computational cost of EQSAM4clim is clearly negligible



for climate applications on architecture such as of the Cy-Tera cluster (Intel Westmere X5650 processors, 2 hexa-core sockets per node). But the differences depend on the system and its usage and are generally smaller on pure scalar architectures. On typical vector machines, however, these differences can significantly increase, since the optimization of a short code can be much more effective. For instance, for the previous supercomputer system at the German Climate Research Center (DKRZ,

www.dkrz.de), the gain in CPU time has been about an order of magnitude. The fraction of the total EMAC CPU burden for a 2 months simulation was for ISORROPIA II about 20 %, while EQSAM4clim only contributed less than 2 % (both on 128 CPUs @ "Blizzard", i.e., IBM Power6 and measured with SCALASCA, http://www.scalasca.org/).

EQSAM4clim has the advantage of being a short fortran 90 code with approximately 850 lines, including comments (or about 8 pages, see Appendix of Metzger et al. (2016b)). For comparison, ISORROPIA II roughly counts 36,300 lines (or

10 approx. 360 pages). This is about 1/3 of the entire source code of the EMAC underlying climate code (ECHAM5.3.02), which has about 119,900 lines of f90 code (also including comments). It should be emphasized that ISORROPIA II was developed for air quality rather than climate modeling, and we offer EQSAM4clim as an alternative for computationally demanding climate simulations.

**Appendix C: Code availability**

EQSAM4clim is freely available for research and non-commercial applications. For commercial applications a special licensing applies. For both cases, please contact the author (swen.metzger@researchconcepts.io). The Modular Earth Submodel System (MESSy) is continuously further developed and applied by a consortium of institutions. The usage of MESSy and access to the source code is licensed to all affiliates of institutions which are member of the MESSy Consortium. Institutions can be member of the MESSy consortium by signing the MESSy Memorandum of Understanding. More information can be

found on the MESSy Consortium Website (http://www.messy-interface.org/).





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





**Table 1.** Modeling systems that provide an option to use EQSAM and/or ISORROPIA. References are given for certain model versions: EQSAM_v03d (EQ1, Metzger et al. (2002b)), EQSAM2 (EQ2, Metzger et al. (2006)), EQSAM3 (EQ3, Metzger and Lelieveld (2007)), EQSAM4clim (EQ4c, Metzger et al. (2016b)) and/or ISORROPIA-I (ISO1, Nenes et al. (1998)), ISORROPIA-II (ISO2, Fountoukis and Nenes (2007)). URL: CAMx – http://www.camx.com; CHIMERE – http://www.lmd.polytechnique.fr/chimere/; EMAC – http://www.messy-interface.org/; EMEP – http://www.emep.int; GEOS – https://gmao.gsfc.nasa.gov/GEOS; LOTOS-EUROS – https://lotos-euros.tno.nl; Meso-NH – http://mesonh.aero.obs-mip.fr/; NASA GISS – https://www.giss.nasa.gov; POLYPHEMUS – http://cerea.enpc.fr/polyphemus; RACMO – https://www.knmi.nl/; WRF – https://www.mmm.ucar.edu/weather-research-and-forecasting-model.

| Modeling System | Model | Reference |
|---|---|---|
| CAMx | ISO1 | Koo et al. (2009), CAMx User's Guide Version 6.40 (Environ, 2016) |
| CHIMERE | EQ1 / ISO1 | Bessagnet et al. (2004), de Meij, A. (2009) |
| EMAC/GMXe | EQ4c | Metzger et al. (2016b), Metzger et al. (2016a), Abdelkader et al. (2017) |
| EMAC/GMXe | ISO2 | de Meij et al. (2012), Karydis et al. (2016), Pozzer et al. (2017) |
| EMAC/MADE3 | EQ1 | Lauer et al. (2005), Kaiser et al. (2014) |
| EMEP | EQ1 / ISO1 | Simpson et al. (2003), Simpson et al. (2012) |
| GEOS-5 | EQ1 | Darmenov et al. (2016), https://bit.ly/2FaO7E1 |
| Meso-NH | EQ1 / ISO1 | Lac et al. (2018), General docu of Meso-NH v5.1 (https://bit.ly/2pJDe2c) |
| NASA/GISS | EQ1 / ISO2 | Bauer et al. (2007a), Bauer et al. (2007b), Bian et al. (2017) |
| WRF/POLYPHEMUS | EQ1 / ISO2 | Zhang et al. (2013); Polyphemus 1.6 User's Guide https://bit.ly/2DVNpG4 |
| RACMO-LOTOS-EUROS | EQ1 | Van Meijgaard and KNMI. (2008), Manders et al. (2011) |
| TM3/TM5 | EQ1 / ISO2 | Metzger et al. (2002a), Dentener et al. (2002), de Meij et al. (2006) |
| CAMx | EQ4c | Under evaluation (Koo et al., 2018), CAMx User's Guide Version 6.45 |
| EMEP | EQ4c | Implementation in progress (inclusion foreseen in Report 2018) |

**Table 2.** Parameters for the different aging levels shown in Table 4 (Sec. 4.2). $\nu_{bulk}$ $[-]$ denotes the bulk water uptake coefficient, $RHD_{bulk}$ [%] the bulk water uptake threshold and $MF_{bulk}$ [%] the mass fraction used for chemical aging of the bulk aerosol species. The main reagent that is assumed to determine the aging (through implicit coating and water uptake) is included below the bulk species. The values have been empirically determined by numerous model applications and a very comprehensive model evaluation by the constraint to yield the best agreement of our EMAC version with independent model results and various observations. Key results of this evaluation cycle are shown in Sec. 3, additional results will be presented separately.

| Bulk compound with main reagent | BC $NH_4NO_3$ \|$NH_4HSO_4$ | OC $(NH_4)_2SO_4$\|$NH_4HSO_4$ | DU $Ca(Cl)_2$ \| $Ca(NO_3)_2$ | SS NaCl \| NaCl |
|---|---|---|---|---|
| Aging case | 50 \| 90% | 50 \| 90% | 50 \| 90% | 50 \| 90% |
| $\nu_{bulk}$ | 1.051 \| 1.254 | 1.275 \| 1.254 | 2.025 \| 1.586 | 1.358 \| 1.358 |
| $MF_{bulk}$ | 50 \| 90 | 50 \| 40 | 75 \| 90 | 100 \| 50 |
| $RHD_{bulk}$ | 60 \| 40 | 80 \| 90 | 28 \| 49 | 50 \| 75 |





**Table 3.** EMAC tracer statistics for the year 2005 and 189 stations based on 5 hourly model output. Simulations based on ISORROPIA II (ISO2) and EQSAM4clim (EQ4c) (identical EMAC setup).

| | STATION MEAN | | RMSE | CORR | MBE |
|---|---|---|---|---|---|
| | ISO2 | EQ4c | – | – | – |
| PM | 58.05±193.45 | 57.23±193.03 | 3.64 | 1.00 | -0.82 |
| DU | 41.91±192.84 | 41.75±192.34 | 3.41 | 1.00 | -0.17 |
| SS | 6.83± 8.47 | 6.37± 7.78 | 0.93 | 1.00 | -0.45 |
| OC | 2.32± 1.94 | 2.33± 1.94 | 0.07 | 1.00 | 0.01 |
| BC | 0.45± 0.56 | 0.45± 0.56 | 0.01 | 1.00 | -0.00 |
| $H_2O$ | 14.48± 13.71 | 13.53± 13.07 | 2.32 | 0.99 | -0.96 |
| $NO_3^-$ | 1.26± 1.02 | 1.16± 0.95 | 0.30 | 0.96 | -0.10 |
| $SO_4^{2-}$ | 2.25± 1.53 | 2.40± 1.66 | 0.32 | 0.99 | 0.15 |
| $H_2SO_4$ | 0.02± 0.03 | 0.02± 0.03 | 0.00 | 1.00 | -0.00 |
| $HSO4^-$ | 0.22± 0.47 | 0.12± 0.27 | 0.24 | 0.99 | -0.10 |
| $Ca^{2+}$ | 2.25± 10.28 | 2.24± 10.26 | 0.18 | 1.00 | -0.01 |
| $Mg^{2+}$ | 0.19± 0.24 | 0.18± 0.22 | 0.03 | 1.00 | -0.01 |
| $NH_4^+$ | 0.85± 0.71 | 0.81± 0.69 | 0.09 | 0.99 | -0.04 |
| $Na^+$ | 0.66± 0.81 | 0.62± 0.75 | 0.08 | 1.00 | -0.04 |
| $Cl^-$ | 0.64± 0.90 | 0.56± 0.83 | 0.15 | 0.99 | -0.08 |
| $K^+$ | 0.19± 0.12 | 0.19± 0.12 | 0.01 | 1.00 | -0.00 |
| $H^+$ | 0.02± 0.02 | 0.02± 0.02 | 0.01 | 0.93 | 0.00 |
| $OH^-$ | 0.06± 0.09 | 0.06± 0.09 | 0.02 | 0.97 | 0.00 |
| NO | 0.63± 1.09 | 0.62± 1.07 | 0.09 | 1.00 | -0.00 |
| $NO_2$ | 6.00± 6.70 | 5.98± 6.66 | 0.18 | 1.00 | -0.02 |
| $SO_2$ | 3.53± 3.28 | 3.50± 3.25 | 0.13 | 1.00 | -0.03 |
| $HNO_3$ | 1.64± 2.01 | 1.69± 2.05 | 0.21 | 1.00 | 0.05 |
| HCl | 0.20± 0.20 | 0.21± 0.20 | 0.08 | 0.93 | 0.01 |
| $O_3$ | 56.61± 19.34 | 56.41± 19.29 | 0.69 | 1.00 | -0.20 |
| RWETAER | 1.95± 0.17 | 1.95± 0.17 | 0.03 | 0.99 | 0.00 |
| RDRYAER | 1.75± 0.07 | 1.75± 0.06 | 0.01 | 0.98 | 0.00 |
| AERNUMB | 260.36±130.37 | 264.54±132.55 | 21.72 | 0.99 | 4.18 |
| RH | 69.16± 20.81 | 69.20± 20.79 | 0.69 | 1.00 | 0.04 |
| T | 18.94± 28.84 | 18.95± 28.83 | 5.11 | 0.98 | 0.01 |





**Table 4.** Sensitivity runs with different levels of aerosol aging as defined in Sec. 4.2 and Table 2. Note that the key difference between "no aging" and "aging" case is the water uptake of primary particles. It is only considered for the latter case (being based on Sec. 2.4 and 2.5). All cases include the GMXe coating processes (Sec. 2.2) through condensation of gases such as hydrochloric acid, nitric acid, sulfuric acid and ammonia on insoluble particles (mineral dust, black and organic carbon). Additionally, in all cases particles can mix through coagulation, and the formation of semi-volatile salt-compounds such as ammonium nitrate and ammonium chloride, and gas-aerosol partitioning including water uptake (Sec. 2.3), is always applied for compounds in the soluble modes.

| Case | Simulation | Option1 | Option2 | Option3 | Application |
|------|------------|---------|---------|---------|-------------|
|      | "label"    | aerosol water | bulk aging | hysteresis effect | Section |
| 1 | "no aging" | yes | no | no | Sec. 4.1 + S1.3 |
| 2 | "no water" | no | no | no | Sec. 2.5 |
| 3 | 50% aging | yes | 50% | yes | Sec. 4.2 + 3.3 |
| 4 | "90% aging" | yes | 90% | yes | Sec. 2.5 |

**Table 5.** CPU times. EMAC @ 96 CPU cores, Cy-Tera (http://web.cytera.cyi.ac.cy/).

| Simulation | Memory [Gb/node] | CPU-time [h/node] | Wall-time [h] |
|------------|------------------|-------------------|---------------|
| A – ISORROPIAII | 5.713064 | 173:49:49 | 14:31:26 |
| B – EQSAM4clim | 5.751476 | 158:53:35 | 13:16:42 |
| C – EQSAM4clim | 5.756064 | 158:08:04 | 13:12:58 |
| D – none of both | 5.738376 | 153:11:01 | 12:48:10 |
| $A_0$ – ISORROPIAII | 5.748988 | 172:33:56 | 14:25:05 |
| $B_0$ – EQSAM4clim | 5.744580 | 152:24:34 | 12:44:16 |

**Table A1.** List of names and abbreviations.

| Abbreviation | Name |
|--------------|------|
| AERONET | AErosol RObotic NETwork (http://aeronet.gsfc.nasa.gov) |
| AOD | Aerosol Optical Depth (https://earthobservatory.nasa.gov/GlobalMaps/view.php?d1=MODAL2_M_AER_OD) |
| CPU | Computational Performance Unit (https://en.wikipedia.org/wiki/Central_processing_unit) |
| Cy-Tera | The Cyprus Institute high performance computing system (http://web.cytera.cyi.ac.cy/) |
| DKRZ | The German Climate Computing Center high performance computing system (https://www.dkrz.de) |
| EMAC | ECHAM5/MESSy Atmospheric Chemistry-climate model (Joeckel et al. (2005), Joeckel et al. (2010)) |
| EQSAM4clim | Equilibrium Simplified Aerosol Model (Version 4) for Climate Simulations (Metzger et al., 2016b) |
| GMXe | Aerosol microphysics model, Global Modal-aerosol eXtension (Pringle et al., 2010a) |
| ISORROPIA II | Equilibrium Aerosol Model Nenes et al. (1998), Fountoukis and Nenes (2007) |
| MODIS | Satellite observations (http://modis-atmos.gsfc.nasa.gov/) |
| MISR | Global satellite data (https://giovanni.gsfc.nasa.gov/giovanni/) |
| SCALASCA | performance measuring software tool (http://www.scalasca.org/, https://bit.ly/2GF5IoB) |



**AERONET/EMEP stations**

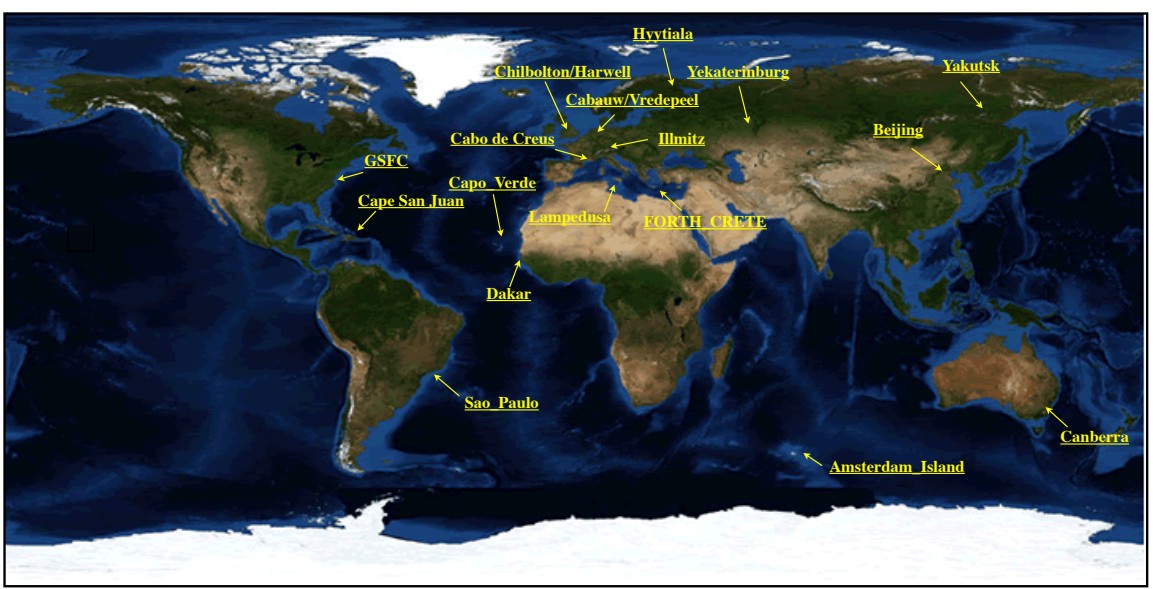

AERONET - http://aeronet.gsfc.nasa.gov/cgi-bin/type_piece_of_map_opera_v2_new          EMEP - http://ebas.nilu.no/default.aspx

| Amsterdam_Island (37S,77E) | Sao_Paulo (23S,46W) | Dakar (14N,16W) | Cabauw (51N,4E) | Lampedusa (35N,12E) | Yekaterinburg (57N,59E) | Beijing (39N,116E) |
|---|---|---|---|---|---|---|
| GSFC (38N,76W) | Cape_San_Juan (18N,65W) | Capo_Verde (16N,22W) | Chilbolton (51N,1W) | FORTH_CRETE (35N,25E) | Yakutsk (61N,129E) | Canberra (35S,149E) |
| EMEP | Cabo de Creus (42N,3E) | Vredepeel (51N,4E) | Harwell (51N,1W) | Cabauw (51N,4E) | Illmitz (48N,16E) | Hyytiala (61N,24E) |

**Figure 1.** Locations of selected AERONET and EMEP stations used in this EMAC evaluation study. The corresponding regions are shown in Fig. S1 (Supplement).





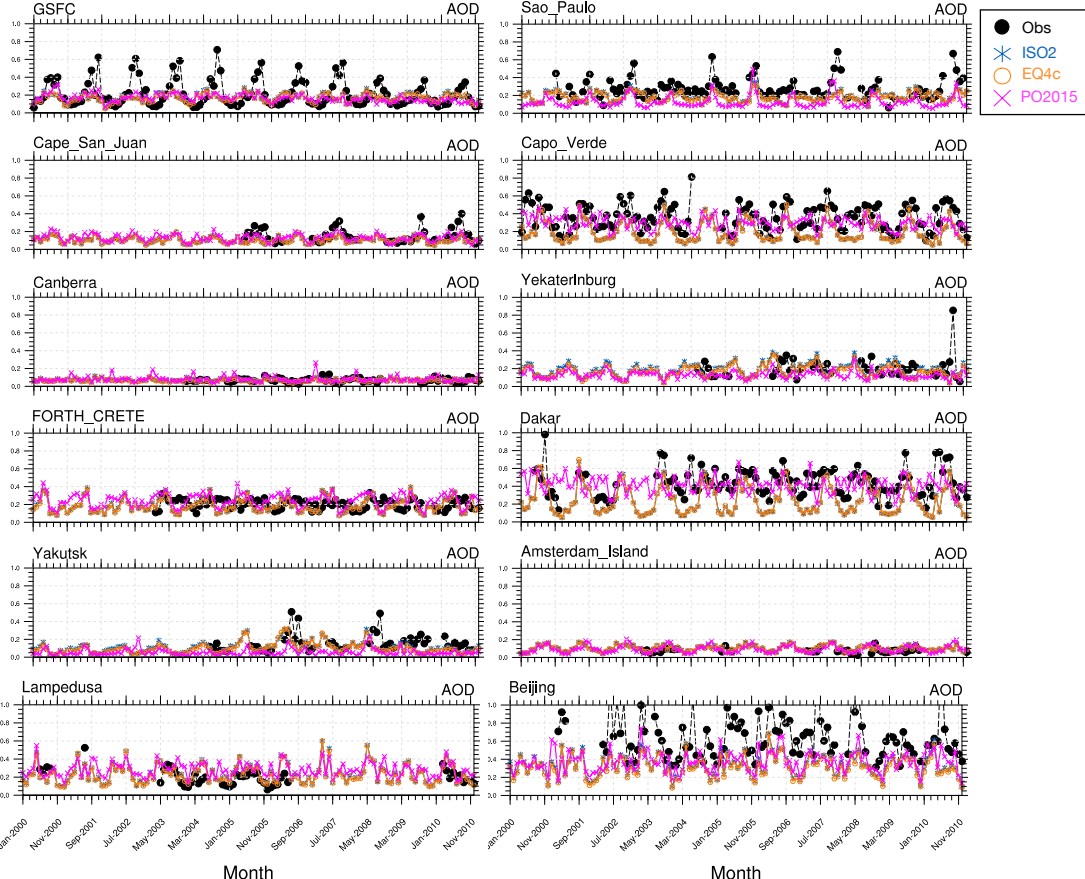

**Figure 2.** Selected AOD time-series for 2000-2010 (monthly means) for the stations shown in Fig. 1, representing all regions of Fig. S1. EMAC results based on ISORROPIA II (ISO2), EQSAM4clim (EQ4c) versus AERONET observations (black circles) and Pozzer et al. (2015) (PO2015). Additionally, scatter plots (Figs. S2– S4) are shown in the Supplement for 537 AERONET stations (Fig. S1).





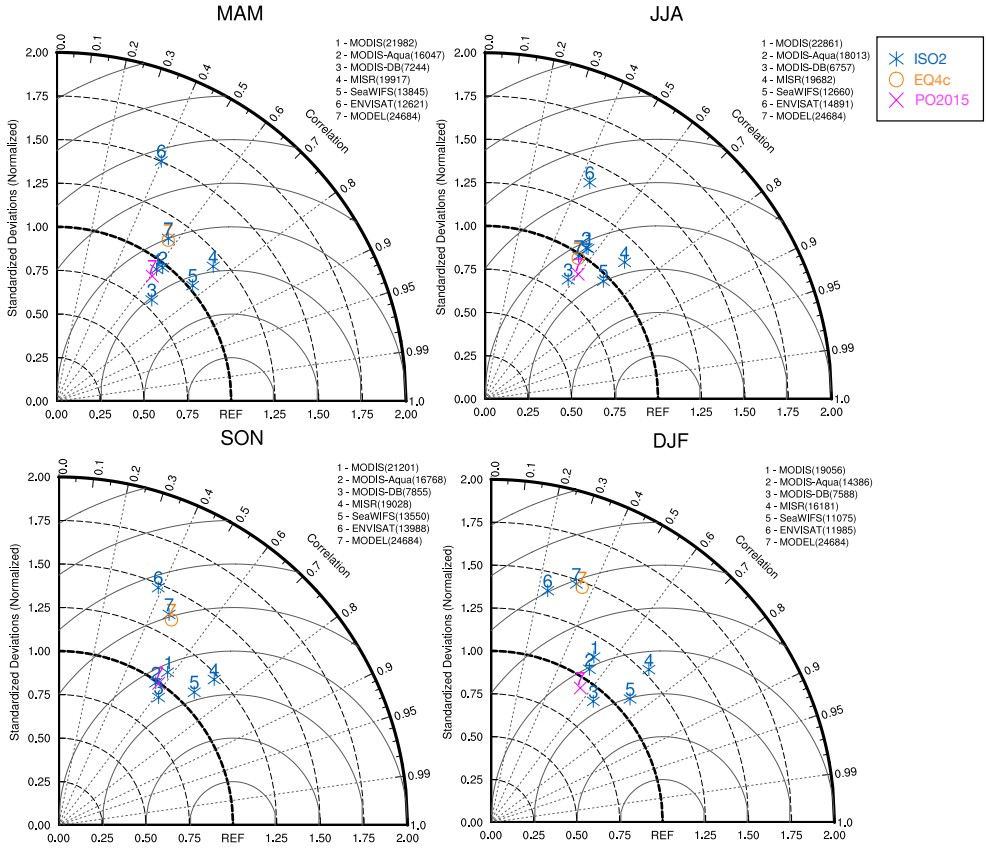

**Figure 3.** Taylor diagram for satellite and model AOD (2000-2010 mean). MODIS (1), MODIS-Aqua (2), MODIS-deep blue (3), MISR (4), SeaWIFS (5), ENVISAT (6) and models (7), i.e., ISORROPIA II (ISO2), EQSAM4clim (EQ4c), Pozzer et al. (2015) (PO2015), versus AERONET observations for the four seasons: spring (MAM), summer (JJA), autumn (SON), winter (DJF). The number of observational points used in the seasonal analysis are shown in parenthesis.



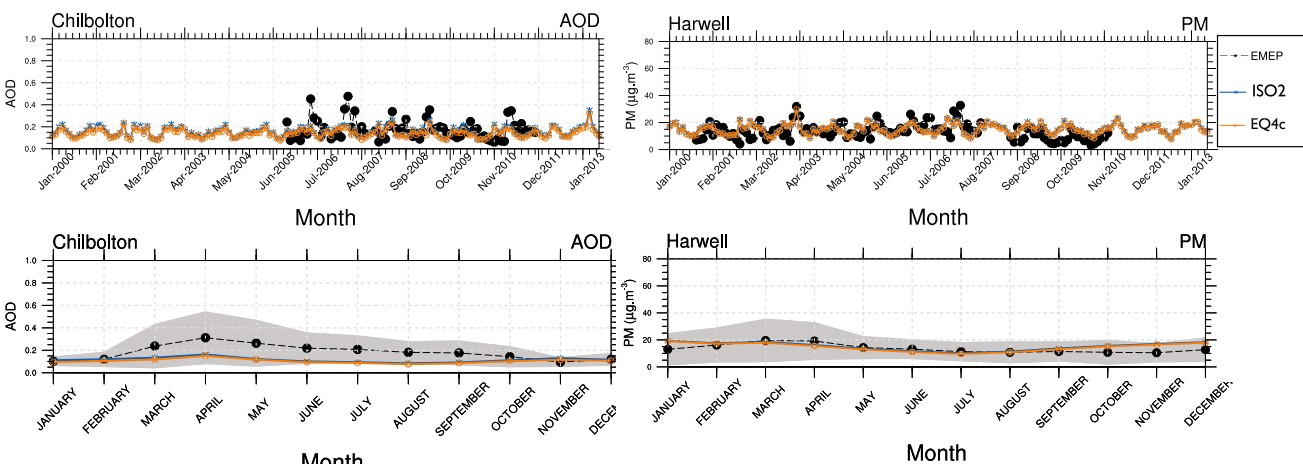

**Figure 4.** AOD and PM time-series for 2000-2013 (monthly means): ISORROPIA II (ISO2) and EQSAM4clim (EQ4c) versus AERONET and EMEP observations (upper panels). The lower panels show the corresponding climatological year for the AOD and PM (14 years average). The two stations Harwell and Chilbolton (United Kingdom) lie well within one model grid box (51N,1W).





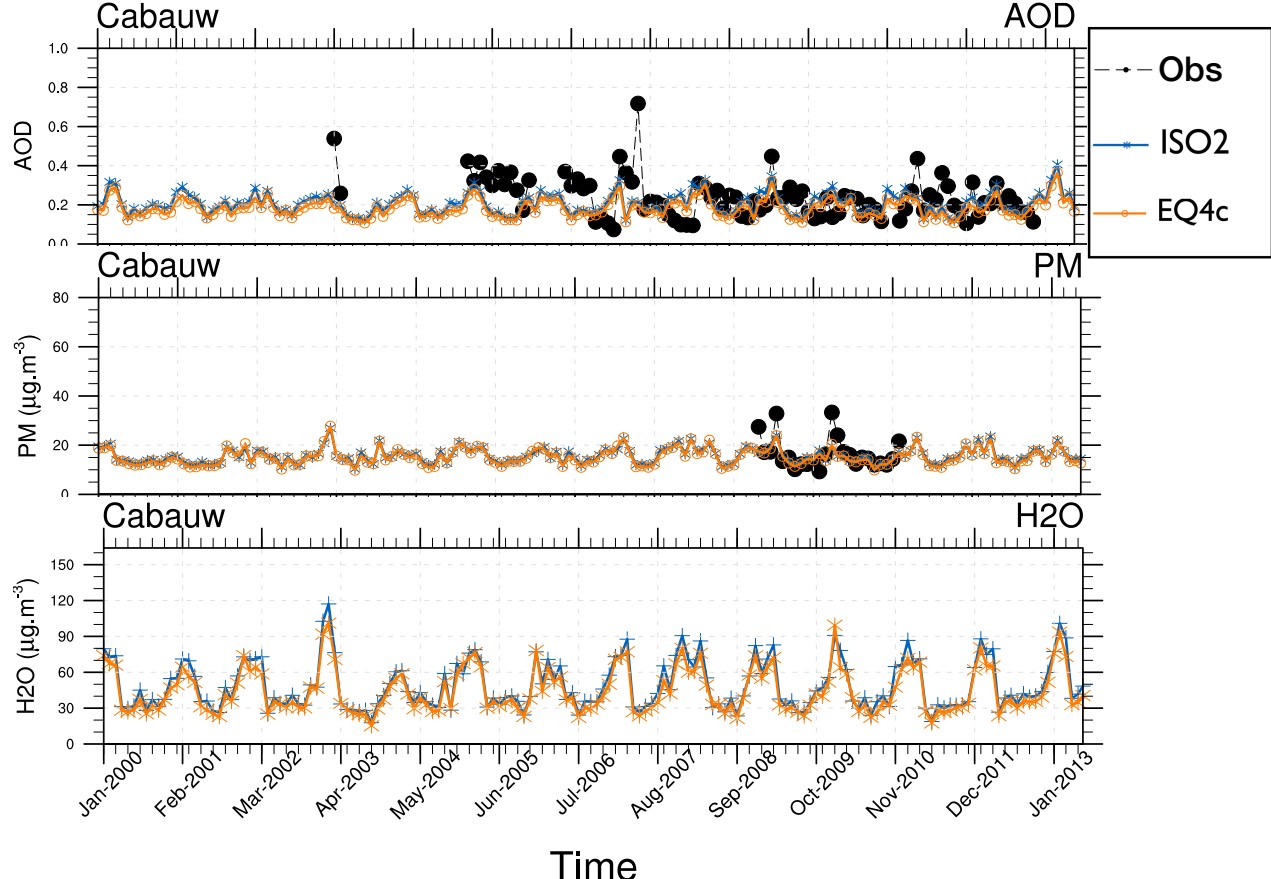

**Figure 5.** AOD (top), total (liquids and solids) particulate matter (PM) (middle), aerosol associated water (bottom) at EMEP station Cabauw for 2000-2013 (monthly means): ISORROPIA II (ISO2), EQSAM4clim (EQ4c), AERONET observations (black circles). Available observations are shown. Additionally, various size-resolved aerosol properties are shown in the Supplement (Figure S5).



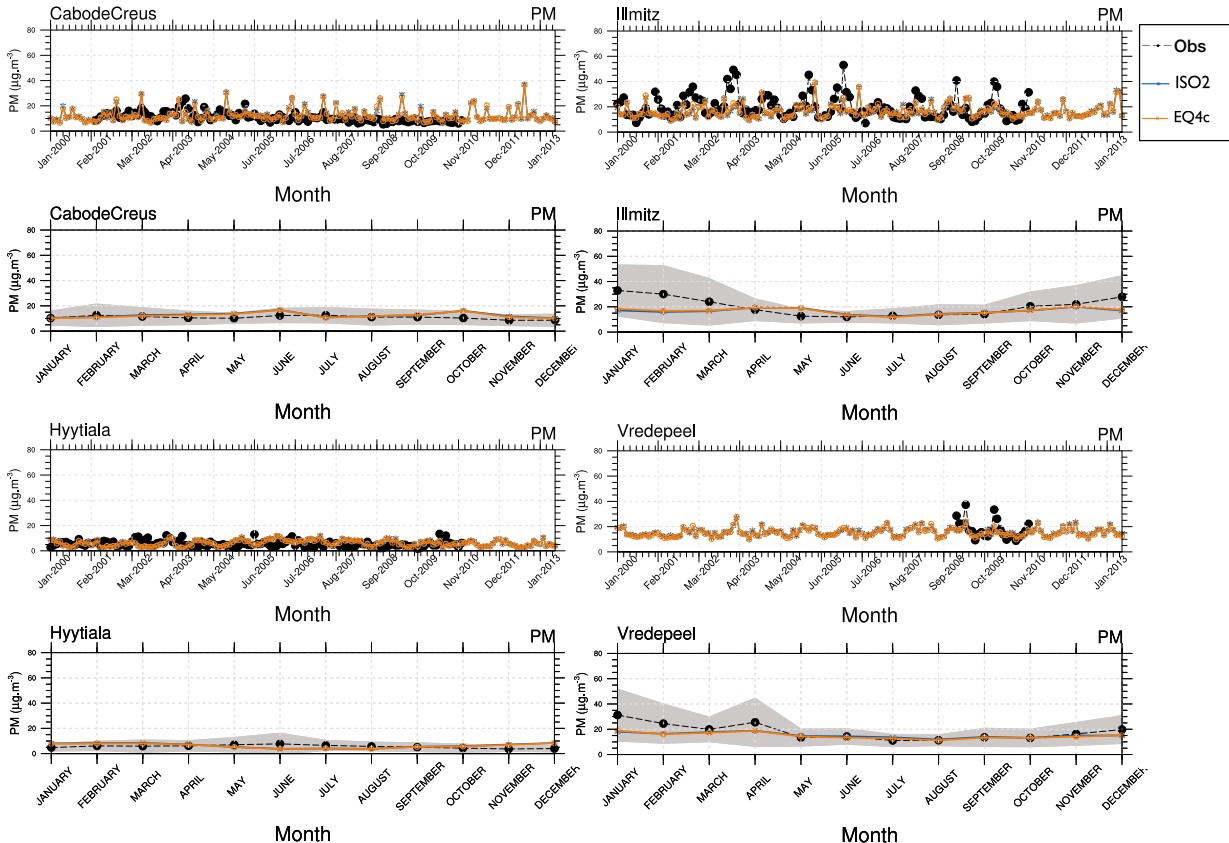

**Figure 6.** Aerosol mass (PM) time-series for 2000-2013 (monthly means): ISORROPIA II (ISO2) and EQSAM4clim (EQ4c) versus EMEP stations, which have long-term observations (top panels). The corresponding climatological year (14 years average) is shown below each time-series.





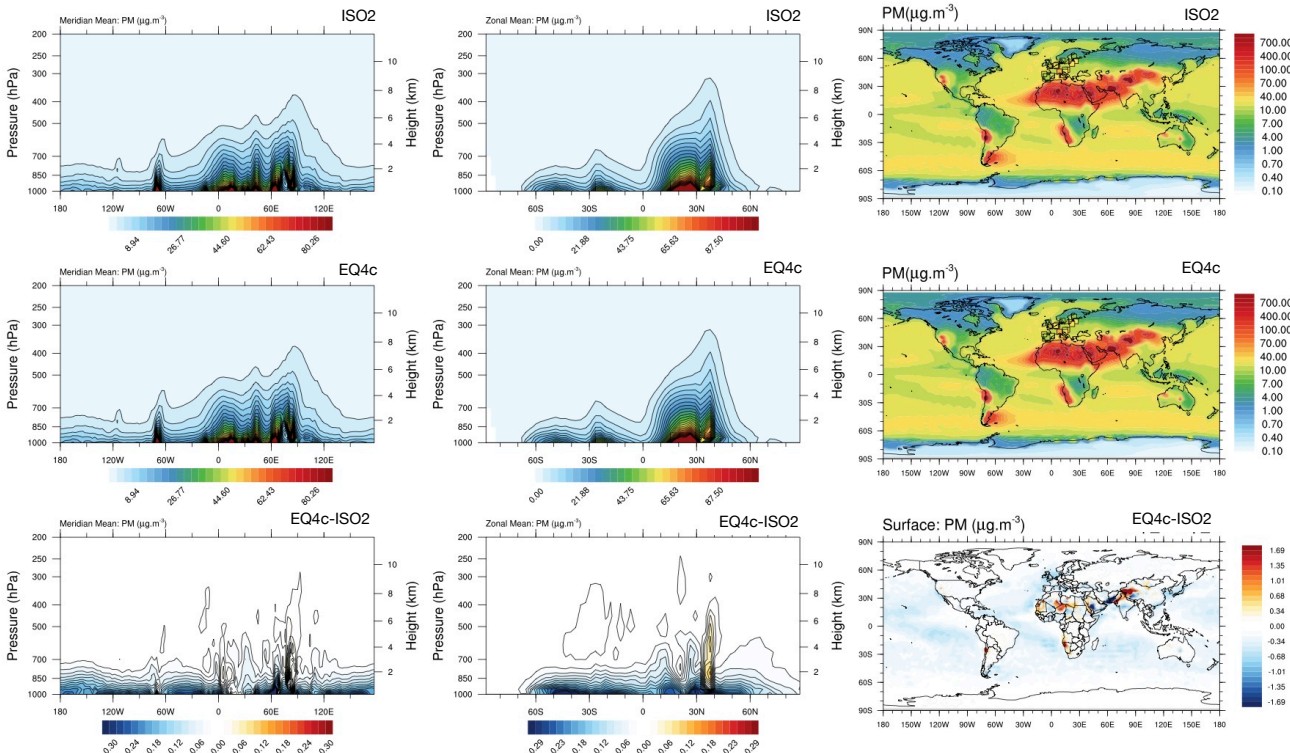

**Figure 7.** Global aerosol distributions of the total (liquids and solids) particulate matter: meridional mean (left column), zonal mean (middle column), atmospheric burden (right column). The EMAC results shown are based on ISORROPIA II (ISO2, top row), EQSAM4clim (EQ4c, middle row), and the corresponding difference between both simulations (EQ4c minus ISO2, bottom row).



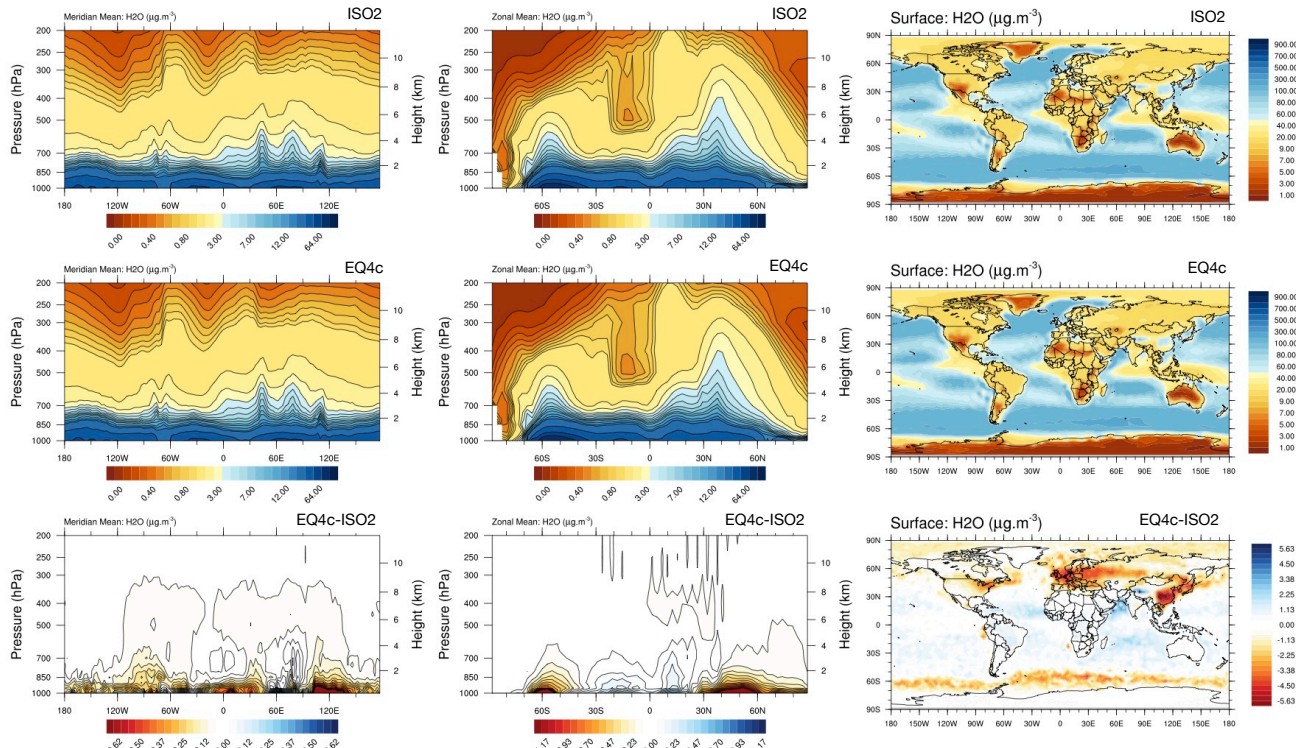

**Figure 8.** Fig. 7 continued for EMAC aerosol associated water (2000-2013 mean).





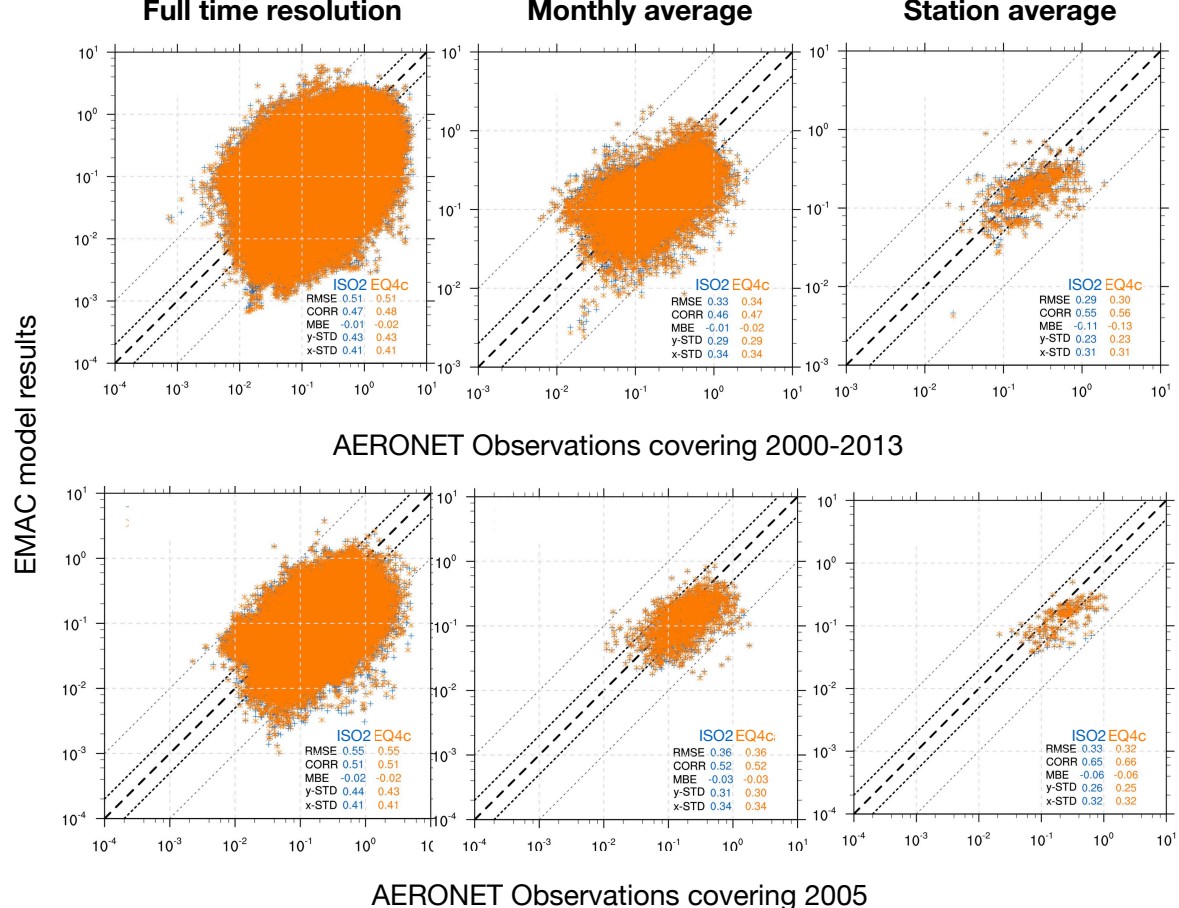

**Figure 9.** EMAC AOD versus AERONET observations for the period 2000-2013 (top) and the year 2005 (bottom). Different time averages are shown for the results of ISORROPIA II (ISO2) and EQSAM4clim (EQ4c) based on 537 AERONET station locations (Figure S1 of the Supplement).



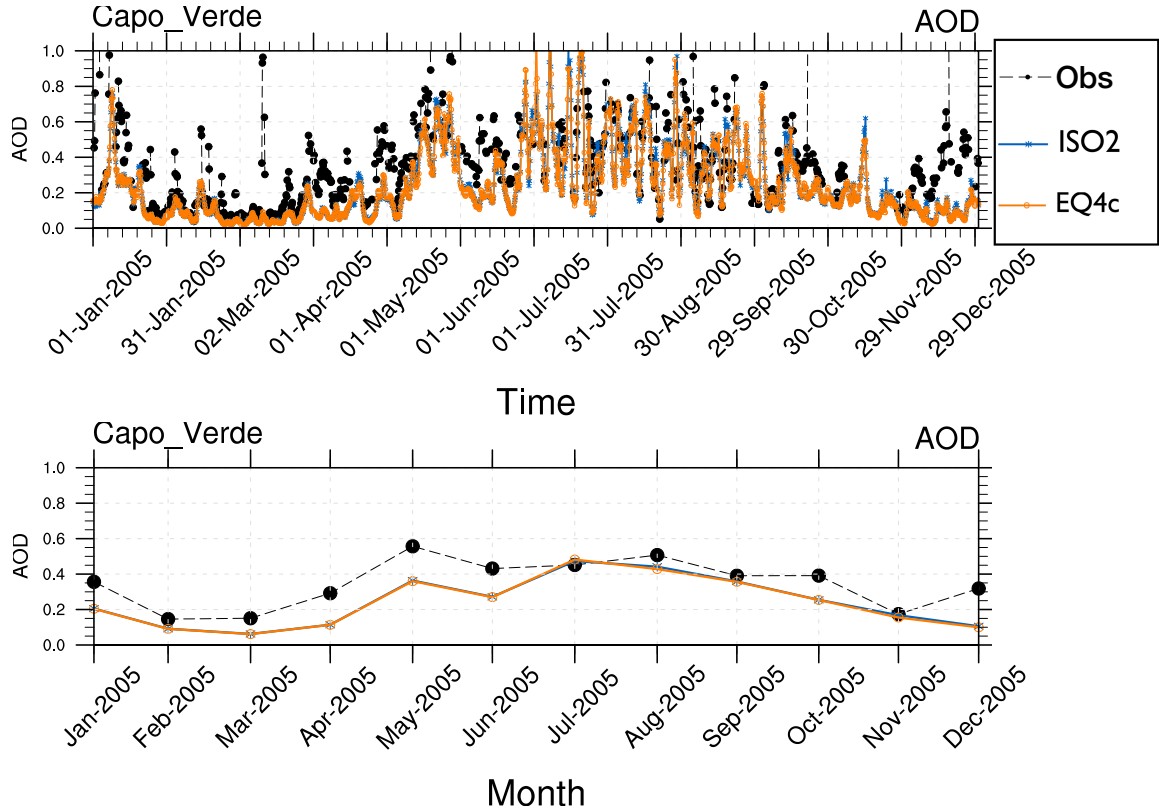

**Figure 10.** EMAC AOD results versus AERONET observation (Obs) at Capo Verde (year 2005); (top) 5 hourly means, (bottom) monthly means; EQSAM4clim (EQ4c) and ISORROPIA II (ISO2).





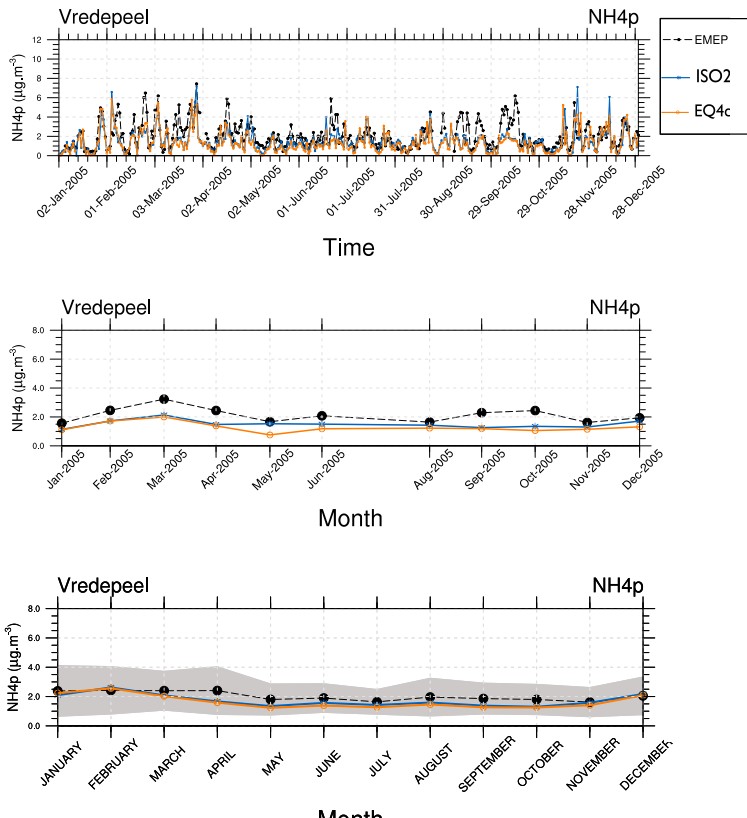

**Figure 11.** Total (liquids and solids) particulate ammonium (NH$_4^+$) at EMEP site Vredepeel for the year 2005. Daily means (top), monthly means (middle), climatological year based on the 14 years monthly mean (bottom). ISORROPIA II (ISO2), EQSAM4clim (EQ4c) versus observations (EMEP).





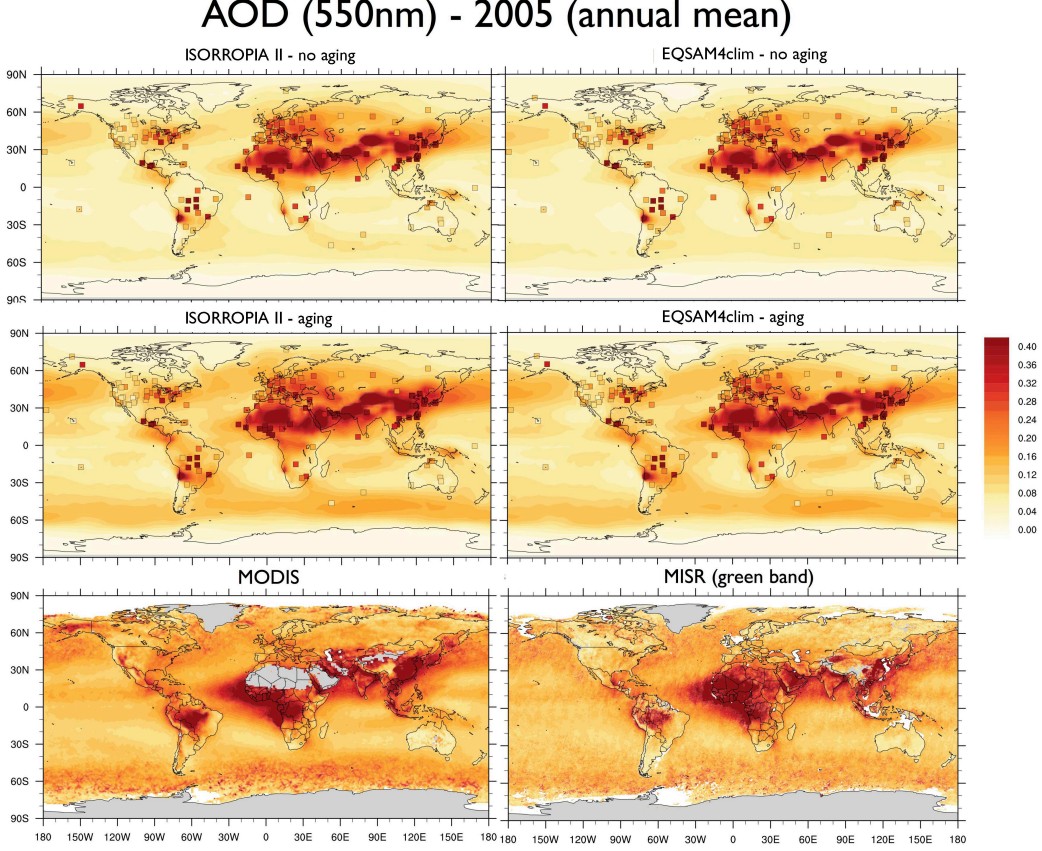

**Figure 12.** EMAC model AOD results for the year 2005 (annual mean) based on ISORROPIA II (left) and EQSAM4clim (right). Upper row: "*no aging*", middle row: "*aging*" case. AERONET ground station observations are included as squares (same color scale). Lower row: Satellite observations by MODIS (left) and MISR (right) (550nm, annual mean 2005). MODIS monitors the ambient AOD from space and provides data over the oceans and, except deserts, also over continents (http://modis-atmos.gsfc.nasa.gov/). The MISR aerosol product is available globally (products can be obtained from http://disc.sci.gsfc.nasa.gov/giovanni).





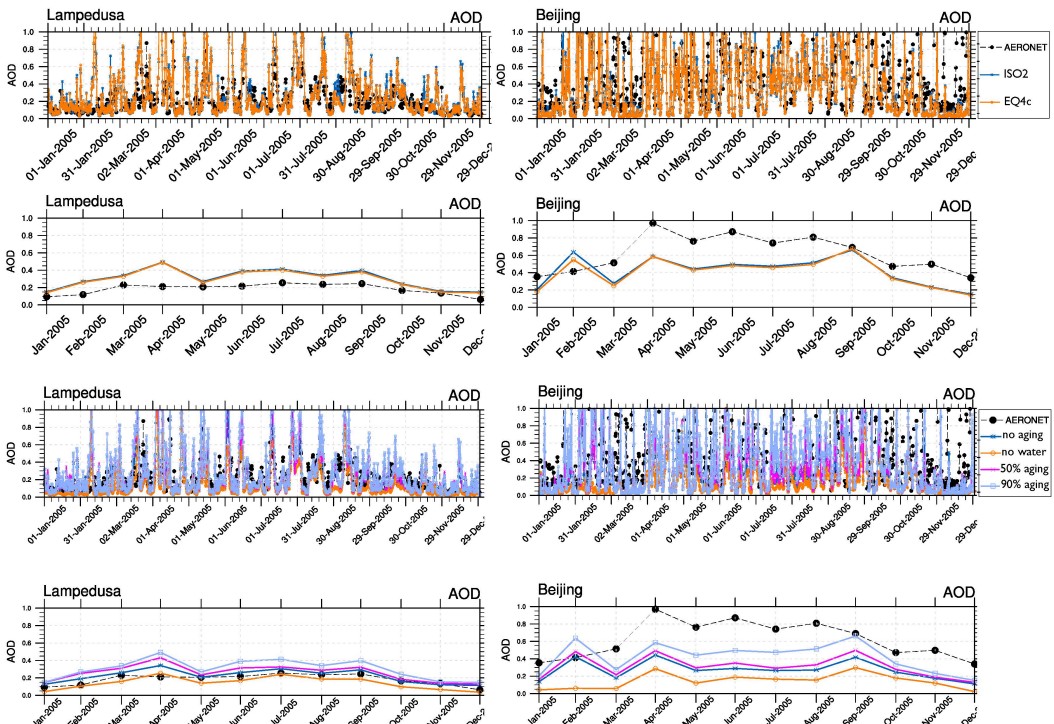

**Figure 13.** EMAC AOD results versus AERONET observations at Lampedusa and Beijing (shown in Fig. 2) for the year 2005. First and third row: 5 hourly means; 2nd and 3rd row: monthly means. Upper two rows show EQSAM4clim (EQ4c) and ISORROPIA II (ISO2), lowest two rows sensitivity of EMAC AOD to different water assumptions considering different EMAC setups; "no aging" (blue stars), "no water" without aerosol water (orange circles), "50% aging" (pink crosses), "90% aging" (light blue squares); see Table 4 and Table 2 (Sec. 4.2). The sensitivity is based on ISORROPIA II.




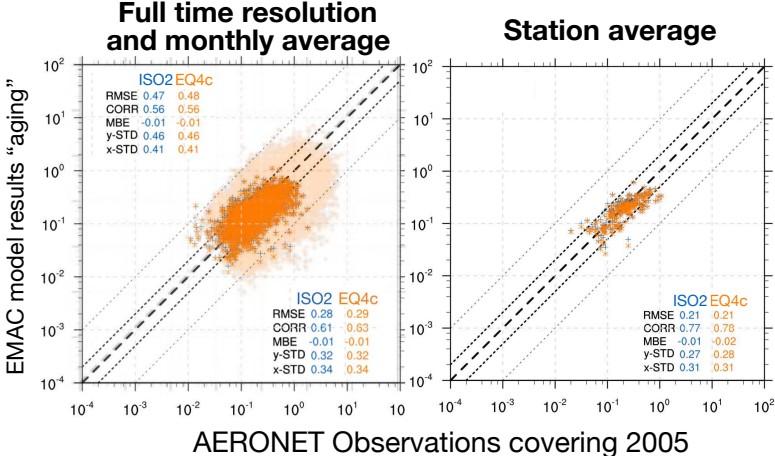

**Figure 14.** EMAC AOD based on the "aging" set-up versus AERONET observations for the year 2005, complementing Fig. 9. Different time averages (full time resolution in light colors with statistics in the upper left corner) are shown for the results of ISORROPIA II (ISO2) and EQSAM4clim (EQ4c) based on 537 AERONET station locations (shown in Figure S1 of the Supplement).




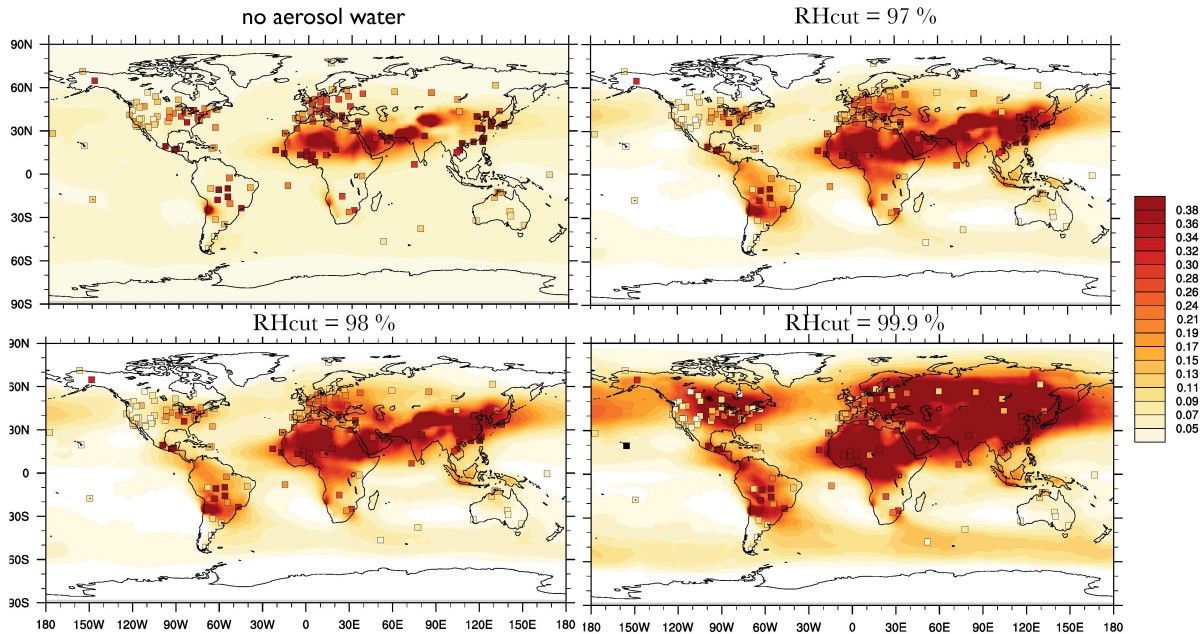

**Figure 15.** Sensitivity of EMAC AOD to different RH cut-offs (see text Sec. 4.3).