# Peer review of "Aerosol water parameterization: long-term evaluation and importance for climate studies"

_Atmospheric Chemistry and Physics, 2018_

## Referee Comment (RC1) · Anonymous Referee #2 · 1 Aug 2018

Metzger et al. present global model simulations of AOD that account for the uptake of particulate water due to inorganic aerosol using EQSAM4clim as well as other methods such as ISORROPIA II and a previous study. The conclusions outlined in the abstract (consistency with ISORROPIA, Pozzer et al. (2015) and this work as well as sensitivity of aerosol water to RH near 100%) seem supported by the figures. However, the manuscript would benefit from clarifying what is meant by "aging" and providing more information on EQSAM4clim.

Major comments:

1. The aging vs no aging cases were not completely clear and largely read as a sensitivity of hysteresis. Is it correct that "aging" is really a hysteresis assumption with increasing deliquescence as a function of age? With age, couldn't particles have

deliquesced or effloresced in their history? How is table 2 to be read in terms of bulk compound and reagent? Is OC assumed to coat ammonium sulfate and bisulfate or vice versa?

2. More documentation on EQSAM4clim (briefly presented) would help the reader. For example, Page 7, line 6 references equation A3 of Metzger et al. 2016b. Can that equation be reproduced here since there is no equation in main text for water uptake coefficient? How does the approach here compare to using kappa hygroscopicity parameters instead (Petters and Kreidinweis 2007 ACP https://www.atmos-chem-phys.net/7/1961/2007/acp-7-1961-2007.html)?

3. Clarification in terms of inorganic aerosol components is needed. How is the elemental speciation of dust and seasalt and other bulk species determined? The speciation is discussed in section 2.4 and Table 2 is referenced, but table 2 isn't a direct mapping of bulk to species. Is Table 2 implying dust is Ca(Cl)2 and Ca(NO3)2 while sea salt is NaCl? Is the composition of bulk dust and seasalt tracked in the model or prescribed? Can nitrate replace chloride in sea salt in the model?

Minor comments:

1. Abstract line 10: Why is it important to reproduce Pozzer et al. 2015? Indicate the domain, evaluation data, or some other characteristic that is being reproduced. Perhaps state something along the lines of "...our EMAC results of aerosol optical depath (AOD) are comparable to independent results obtained for [insert description of domain, time period, or identifier that characterizes usefulness of Pozzer et al. 2015] (Pozzer et al., 2015)..."

2. A few references did not display properly in text (for example: page 4 line 13 should have (Ganzeveld et al., 2006) instead of Ganzeveld et al. (2006); Page 4 line 20 uses "poz")

3. Page 3, line 27 "We conclude with section 5" is not necessary.

4. Page 5, near line 28: What are the default cutoffs for minimum and maximum RH for simulations other than the sensitivities exploring cutoff?

5. Can you assess the potential limitations associated with the lack of water uptake on organic aerosol, the effects of organic aerosol on inorganic partitioning and resulting water uptake, and water uptake and resulting AOD? For example, if organics could increase aerosol water by 40%, what would that mean for AOD?

6. Page 8, line 30-31: An evaluation of aerosol composition and neutralization first could have informed some of the comparisons here. I suspect that the AOD will be less sensitive to model assumptions (e.g. ISORROPIA vs EQSAM4clim) than inorganic aerosol composition will be.

7. Page 8, line 29: Were any observations (AERONET?) hourly? Were those observations averaged to the same 5-hour timescale as the model for comparison or were hourly observations only matched with the model when the observation fell exactly in the middle of the 5-hour model prediction?

8. Page 10, section 3.2 introduces 6 figures in less than 1 page without much guidance for the reader in terms of what to take away. Consider summarizing the message from each figure in section 3.2 as the figure is introduced or moving figures that repeat the same message to the SI.

9. Appendix C: I encourage the authors to consider a version controlled (repository) method of code distribution.

10. Figure 1: could be moved to the SI. Is a reference for the figure needed?

11. Figure 9: Consider a scatter density plot or just a table for this information.

12. Can any lessons be learned where all methods fail to capture AOD (e.g. Figure 12 eastern US, Figure 13 Beijing)?

---

## Author Comment (AC1) · 10 Aug 2018

**Response to Referee 2**

S. Metzger et al.

10 August 2018

We thank the referee for the manuscript review. Please find our point-by-point reply below. Accordingly, the revised MS will include clarifications.

Reply to major comments.

1. *The aging vs no aging cases were not completely clear and largely read as a sensitivity of hysteresis. Is it correct that "aging" is really a hysteresis assumption with increasing deliquescence as a function of age?*

   No, "aging" and "hysteresis" are different assumptions in this work. As stated on page 7, line 15:
   "To distinguish between our EMAC setup that considers the water uptake of normally chemically unresolved particles (SS, DU, OC, BC), we use in our study the label "aging", refering to a chemical "aging" that is used in Sec. 4.2. In contrast, our EMAC setup that omits the chemical "aging" and associated water uptake of bulk aerosols is labeled "no aging" (Sec. 4.1)". The hysteresis assumption comes on top of both (Sec. 2.6), but is negligible in our EMAC set-up compared to the "aging" effect, which is why we have not separated it from the sensitivity analysis. So, the differences in AOD between "aging" and "no aging" are basically caused by the associated water uptake of bulk compounds (SS, DU, OC, BC).

   *With age, couldn't particles have deliquesced or effloresced in their history?*

   This is true and considered here in all of our simulations. Inorganic salt compounds, such as NH4NO3, NH4Cl, CaCl2, etc. that might be formed through the coating of primary particles by gases (default

GMXe ageing) exhibit the full gas-liquid-solid phase partitioning, as described in Sec 2.2 and 2.5.

*How is table 2 to be read in terms of bulk compound and reagent? Is OC assumed to coat ammonium sulfate and bisulfate or vice versa?*

Bulk OC is assumed to be coated in the aging case, but its behaviour in terms of water uptake is considered such if a mass fraction of 50% OC would be ammonium sulfate (with the water uptake parameters given in the first sub-column). Respectively for the 90% case, ammonium bisulfate is assumed (with the water uptake parameters given in the second sub-column; see explanation in Table 2).

2. *More documentation on EQSAM4clim (briefly presented) would help the reader. For example, Page 7, line 6 references equation A3 of Metzger et al. 2016b. Can that equation be reproduced here since there is no equation in main text for water uptake coefficient?*

The equation A3 of Metzger et al. 2016b, which is an inversion of Eq. (5a) of Metzger et al. 2012 (M2012), can be reproduced here with the parameters given in Table 2 (with Ke = 1, A = 1, B = 0; see p7, line 7). As detailed in Sec 2.7 of Metzger et al. 2016b (p7223), the mixed solution aerosol water uptake can be obtained by their Eq. (22), from tabulated single solute molalities, or parameterised based on Eq. (5a) of M2012 (Appendix A2, Eq. A3). The effect of the implicit assumption (Ke = 1, A = 1, B = 0) on the overall bulk water uptake is negligible for our simulations (studied but not shown).

*How does the approach here compare to using kappa hygroscopicity parameters instead (Petters and Kreidinweis 2007 ACP https://www.atmos-chem- phys.net/7/1961/2007/acp-7-1961-2007.html)?*

The EQSAM4clim underlying approach has been compared in detail with other approaches, including Kappa hygroscopicity parameters, in M2012; see e.g. their Fig. 3 and 4. The EQSAM4clim approach is the only approach that enables efficient mixed solution water uptake calculations from both, the deliquescence relative humidity up to supersaturation (Köhler curve).

3. *Clarification in terms of inorganic aerosol components is needed.*

According to Sec 2.3:
"Aerosol thermodynamics is represented by EQSAM4clim (Metzger et

al., 2016b) and ISORROPIA II (Fountoukis and Nenes, 2007). For a consistent model inter-comparison, we limit in this study the gas-aerosol partitioning and associated hygroscopic growth of our EMAC simulations to the inorganic compounds considered by ISORROPIA II." Inorganic aerosol components and their thermodynamic properties used in this study are defined in Table 1 of Metzger et al., 2016 (which was limited already to match the compounds of ISORROPIA II).

According to Sec 2.5: "Our chemical speciation of the primary aerosol emission fluxes is coupled to a chemical aging of bulk species through which salt compounds and associated water can be formed. The chemical aging process is hereby based on explizit neutralization reactions of ions (cations, or anions), which are assigned to the emission fluxes (e.g., $K^+$, $Ca^{2+}$, see Sec. 2.4). Through the reactions of these cations (anions) with aerosol precursor gases, i.e., major oxidation products of natural and anthropogenic air pollution (here $H_2SO_4$, $HNO_3$, $HCl$, $NH_3$, and $H_2O$), various neutralization (salt) compounds can be formed, e.g., potassium sulfate ($K_2SO_4$), potassium bisulfate ($KHSO_4$), potassium nitrate ($KNO_3$), potassium chloride ($KCl$), calcium sulfate ($CaSO_4$), calcium nitrate ($Ca(NO_3)_2$), calcium chloride ($CaCl_2$) and so on for ammonium, sodium and magnesium, see Table 1 of Metzger et al. (2016b). The salts can cause an uptake of water vapor ($H_2O$) at different ambient humidities, with $CaCl_2$ at RHs as low as 28%. All salt solutions are subject to the RH and T–dependent gas-liquid-solid partitioning as described in Sec. 2.3 and 2.6."

*How is the elemental speciation of dust and sea salt and other bulk species determined?*

For the chemical speciation applied in this study we follow our approach introduced with Abdelkader et al 2015 and applied in 2017. "This chemical speciation has been determined such that the model concentrations best match the available EMEP and CASTNET measurement data for the period 2000-2013 (to be published separately)." (Abdelkader et al., 2015, p9176, line 13-16). Publication of the comprehensive model evaluation is foreseen and in progress.

According to Sec 2.2, p5, line 12-13: "For the chemical aging we follow our approach introduced with Abdelkader et al. (2015), which is scrutinzed in Section 4.2."

For clarification, this sentence will be changed to: "For the chemical aging of bulk species we follow our approach introduced with Abdelkader et al. (2015), which is scrutinzed in Section 4.2."

Note, Sec 1, p3: "To scrutinize the importance of aerosol water for climate applications, we evaluate the AOD calculations of EQSAM4clim and ISORROPIA II on climate time-scales. For this we extend the model evaluation of (Metzger et al., 2016b) by using the comprehensive chemistry-climate and Earth System model EMAC in a similar setup as applied in our studies on (I) the dust-air pollution dynamics over the eastern Mediterranean (Abdelkader et al., 2015), (II) the sensitivity of transatlantic dust transport 20 to chemical aging and related atmospheric processes (Abdelkader et al., 2017), and (III) the comparison of the Metop PMAp2 AOD products using model data (EUMETSAT ITT 15/210839, Final Report, Metzger et al. (2016a)). These studies employ a highly complex chemistry setup, particularly with respect to the gas-and aqueous phase chemistry and the associated chemical aging of primary aerosols. Since all three studies revealed the importance of chemical aging of primary dust particles for the calculation of the AOD, due to the regionally amplifcation by the aerosol water uptake, its important to evaluate the aerosol water parameterizion also on climate time-scales."

*The speciation is discussed in section 2.4 and Table 2 is referenced, but table 2 isn't a direct mapping of bulk to species. Is Table 2 implying dust is Ca(Cl)2 and Ca(NO3)2 while sea salt is NaCl?*

No., Table 2 gives the fraction of dust that is treated as Ca(Cl)2 (or Ca(NO3)2) for the 50% (or 90 %) aging case (being relevant only for bulk water uptake calculations). Respectively, the same is true for sea salt, OC /BC. Note that this DU fraction is not really chemically resolved and transported as e.g., Ca(Cl)2, so the overall aerosol composition remains indeed unchanged. This is different to our normal (default) GMXe aging, which is considered in all simulations; see p7, line 18-20:

"Independent of this "aging" label, all our EMAC simulations consider a comprehensive treatment of the chemical aging of the non-bulk aerosol emission fluxes, which is part of our GMXe aerosol dynamical treatment Sec. 2.2. The chemical aging includes the dynamically

limited condensation of aerosol precursor gases on primary aerosol particles. Our primary aerosol particles are emitted in the insoluble modes and, depending on the coating level (i.e., the amount of gases condensed on the insoluble particles), they are transferred to the soluble modes. But only the chemically identified compounds of the soluble modes (aitken, accumulation and coarse mode) are subject to the water uptake calculations by either EQSAM4clim or ISORROPIA II by our "no aging" set-up."

In contrast we note for the bulk "aging case": "The required parameters for OC/BC, SS and DU used in our sensitivity study (Sec. 4) to scrutinze the bulk water uptake are given in Table 2 and described in Sec. 2.5" (p, line).

According to Sec 2.5, p7, line 5-10: "To calculate the bulk water uptake, we use the EQSAM4clim parameterizations (introduced by Metzger et al. (2012)) and solve a bulk solute molality using Eq. A3 of Metzger et al. (2016b). For the sake of simplicity, we neglect the Kelvin-term ($Ke = 1$, $A = 1$, $B = 0$) and further assume that the water uptake of the bulk compounds can be described by a mean value, for which we can use our single coefficient $\nu_i$. We further assume a single chemical reagent to be representative for the bulk water uptake due to chemical aging of the bulk aerosol mass, but we only calculate bulk water uptake if the RH exceeds a certain threshold. This "aging" proxy is given in Table 2 together with the required parameters for our "aging" setup used in Sec. 4.2."

*Is the composition of bulk dust and seasalt tracked in the model or prescribed?*

The composition of bulk dust and sea salt is tracked, but the fraction of chemical speciation for the bulk water uptake (Table 2) is prescribed. The actual composition is calculated online.

According to Sec 2.4, p6, line 13-20: "We assign ions to the bulk emission fluxes of primary aerosols by using the major cations $Na+$, $K+$, $Ca2+$, $Mg2+$ and anions $SO42-$, $Cl-$. Our concept of chemical speciation was originally developed as part of GMXe by Metzger and Lelieveld (2007) to extend the aerosol water uptake calculations to the so far chemically unresolved bulk aerosol mass. Thus, for bio-mass burning OC and BC aerosols, we consider the potassium cation ($K+$) as a key

reagent (proxy) for the water uptake thermodynamics (Sec. 2.3). For DU, we respectively consider as a chemical aging proxy the calcium cation (Ca2+), while we resolve the sea salt emission fluxes in terms of the sea water composition, considering the major cations Na+, K+, Ca2+, Mg2+ and anions Cl- and SO42-. Our emission fluxes of primary sea salt and dust particles are calculated online, in feedback with the EMAC meteorology and radiation computations."

According to Sec 2.5, p7, line 1-5: "For H2O and each cation and anion, a chemical tracer is assigned such that they undergo all aerosol microphysics and thermodynamic processes for their respective GMXe aerosol mode(s) (Sec. 2.2). Through this tracer coupling, each salt compound can alter the subsequent AOD calculations in our EMAC version, most noticeably through an associated aerosol water uptake."

*Can nitrate replace chloride in sea salt in the model?*

Yes, this and much more is the default in all of our simulations and has been detailed by Metzger et al., 2006, 2007, 2012, 2016a,b and Abdelkader et al., 2015 and 2017. We assume these studies to be known.

Reply to minior comments.

1. *Abstract line 10: Why is it important to reproduce Pozzer et al. 2015?. Indicate the domain, evaluation data, or some other characteristic that is being reproduced. Perhaps state something along the lines of ?...our EMAC results of aerosol optical depath (AOD) are comparable to independent results obtained for [insert description of domain, time period, or identifier that characterizes usefulness of Pozzer et al. 2015] (Pozzer et al., 2015). . .?*

   To mention Pozzer et al. 2015 in the abstract is not particularly important. We will change the sentence (p1, line 9-10) to: "... our EMAC results of the aerosol optical depth (AOD) are comparable to modeling results that have been independently evaluated for the period 2000-2010".

2. *A few references did not display properly in text (for example: page 4 line 13 should have (Ganzeveld et al., 2006) instead of Ganzeveld et al. (2006); Page 4 line 20 uses ?poz?)*

   Thanks. This oversight will be fixed.

3. *Page 3, line 27 ?We conclude with section 5? is not necessary.*

   The sentence will be omitted.

4. *Page 5, near line 28: What are the default cutoffs for minimum and maximum RH for simulations other than the sensitivities exploring cut-off?*

   Most of our EMAC simulations use a cutoff of (maximum) RH=95 or 98%. There is no minimum RH by default. In our simulations the minimum RH is determined automatically by the aerosol composition, i.e., by the single solute or mixed solution deliquescence RH (see Sec 2.6 of Metzger et al., 2016).

5. *Can you assess the potential limitations associated with the lack of water uptake on organic aerosol, the effects of organic aerosol on inorganic partitioning and resulting water uptake, and water uptake and resulting AOD? For example, if organics could increase aerosol water by 40%, what would that mean for AOD?*

   Actually, this is shown by our sensitivity analysis and it was our intension to assess this type of question. For instance, Fig. 13 shows the results of different aging assumptions. Although we do not explicitly treat organic aerosols, the 50 and 90% aging case also include water uptake of organic aerosols through our consideration of OC bulk mass (with the parameters given in Table 2). Clearly, only certain regions are dominated by organic aerosols and the water uptake of organic aerosols is usually much less than those of the inorganic counterparts (if normalized to the aerosol mass). Nevertheless, certain regions such as Beijing can be dominated by organic aerosols and the effect on AOD can be significant as shown by Fig. 13 – compare "no water" without aerosol water (orange circles), "50% aging" (pink crosses), "90% aging" (light blue squares) for the monthly means, lowest right panel.

6. *Page 8, line 30-31: An evaluation of aerosol composition and neutralization first could have informed some of the comparisons here. I suspect that the AOD will be less sensitive to model assumptions (e.g. ISORROPIA vs EQSAM4clim) than inorganic aerosol composition will be.*

   Yes, we agree, but the sensitivity of the inorganic aerosol composition to model assumptions (e.g. ISORROPIA vs EQSAM4clim) is presented

in the Supplement of this work (see S1.3, Fig. S6-S20). The extension of our study to a more in-depth evaluation of the underlying aerosol composition and neutralization level is part of the PhD Thesis of Abdelkader et al., 2015 (available from the The Cyprus Institute, or directly from the authors of this study). The key finding of that study support the results shown in this work, though being based on a more comprehensive analysis of the aerosol composition and neutralization levels with a more in depth comparison with measurements.

7. *Page 8, line 29: Were any observations (AERONET?) hourly? Were those observations averaged to the same 5-hour timescale as the model for comparison or were hourly observations only matched with the model when the observation fell exactly in the middle of the 5-hour model prediction?*

   Observations were averaged to the same 5-hour timescale as the model results. We have compared our EMAC results with 1 hourly AERONET observations and various satellite data in Metzger, et al., 2016a: Comparison of Metop PMAp Version 2 AOD Products using Model Data, Final Report EUMETSAT ITT 15/210839, http://bit.ly/2Epxf9b, which also details the interpolation procedure in time and space.

8. *Page 10, section 3.2 introduces 6 figures in less than 1 page without much guidance for the reader in terms of what to take away. Consider summarizing the message from each figure in section 3.2 as the figure is introduced or moving figures that repeat the same message to the SI.*

   The results of each figure will be summarized as the figure is introduced.

9. *Appendix C: I encourage the authors to consider a version controlled (repository) method of code distribution.*

   A version controlled repository of code distribution is foreseen.

10. *Figure 1: could be moved to the SI. Is a reference for the figure needed?*

   We prefer to have Fig 1 in the main text for an easier overview of the station locations used in the main work. The appendix additionally shows the locations of all stations used in our model evaluation.

11. *Figure 9: Consider a scatter density plot or just a table for this information.*

We prefer to keep Fig 9, since it nicely illustrates and summarizes the effect of time averaging that can't be better represented in our opinion.

12. *Can any lessons be learned where all methods fail to capture AOD (e.g. Figure 12 eastern US, Figure 13 Beijing)?*

    Yes, according to our experience, most likely the aerosol-cloud interaction needs to be improved by considering a mass conservative approach as suggested by Metzger and Lelieveld, 2007. However, this is beyond the scope of this work and subject for a follow-up study.

---

## Referee Comment (RC2) · Anonymous Referee #1 · 14 Sep 2018

The stated goal of this manuscript is to evaluate the importance of aerosol water for AOD calculations in a long-term (∼one decade) climate simulation using the EQuilibrium Simplified Aerosol Model V4 for climate (EQSAM4clim). The authors argue this modeling parameterizaition is computationally efficient and does not degrade model performance when evaluated with AOD (e.g., AERONET and EMEP) measurements in climate modeling applications relative to a more explicit approach, i.e., ISORROPIA. The implemented EQSAM4clim in the climate model, EMAC, compares reasonably with other AOD comparison results in the literature.

I find the results are supportive of the abstract's stated main conclusion, that aerosol water is important for climate applications. The paper needs work prior to final publication and some results contradict the literature. In Figure 8 the authors find that aerosol

water mass concentrations are highest in the western desert of the U.S. Liao and Seinfeld 2005 (https://agupubs.onlinelibrary.wiley.com/doi/abs/10.1029/2005JD005907) and Carlton and Turpin 2013 (https://www.atmos-chem-phys.net/13/10203/2013/) find it is highest in the eastern U.S. where sulfate mass concentrations and RH are higher. Can the authors provide a context for this discrepancy? The authors stress the importance of 'aging' in their manuscript and this is not well described. I think they mean changing particle hygroscopicity with time but this is not clearly stated. Also, 'aging' is a not a precise term. Do they mean increased oxidation due to longer OH exposure in the atmosphere, and the subsequent changing chemical composition important water uptake? The authors do not make a compelling case in the introduction for their work and I found this confusing. Below I provide comments that think help address my concerns and I hope the authors find them useful.

The introduction is not directly linked to the premise that aerosol water is crucial. I find it difficult to understand why the introduction starts with the importance of desertification and subsequent dust emissions to properly describe AOD when their title and abstract focus on aerosol water. Do the authors mean to say that even in arid regions, AOD is not properly described in models unless water uptake is considered? That is a compelling argument and would help to connect the introductory desert discussion with aerosol water.

Sentence 1: "providing realistic projections of climate change is one of the most difficult tasks of climate modelers..." I would state "Providing realistic projections of climate change is difficult due to many unknowns and large uncertainties . . . " As written the sentence is awkward, the authors seem to say, conducting climate simulations is the hardest thing for climate modelers to do.

The authors might not be aware of this paper using actual measurements of particle-phase ions and meteorology coupled with ISORROPIA-II to calculate aerosol water to better connect surface particle mass measurements to satellite AOD by Nguyen in Geophys. Res. Letts.:

[Figure]

https://agupubs.onlinelibrary.wiley.com/doi/10.1002/2016GL070994 I think it would help with their argument to link the importance of aerosol water with AOD and then subsequently radiaitvie forcing calculations important for climate modeling efforts.

Page 4, Line 13: is dry deposition based only properties of the surface? Do different chemical species all deposit at the same rate?

Page 4, Line 9/25: Why are some subroutines listed together, while others are separated out? For example CLOUD, CVTRANS, JVAL, TROPOP, H2O, ORBIT, and RAD are listed below in a similar fashion.

Page 4, Line 16/17: what do "...water isoprene concentration" and "methanol water deposition..."? Perhaps the authors mean isoprene concentration in ocean water? Do they mean dry deposition of methanol to water? I read this sentence multiple times and I am still not sure.

Page 4, Line 20: Does "Our chemical mechanism for the troposphere is similar to the one used in poz" mean the mechanism is the same as used in Pozzer et al., 2006, cited earlier? Sometimes the authors write "poz" and "Pozzer". I am not sure if they mean the same thing.

Page 5, last sentence and continuing to the next page: "It was shown by Metzger et al. (2016b) that the i -approach allows to analytically solve the gas-liquid-solid partitioning and the mixed solution water uptake by eliminating the need for numerical solutions . . ." Is the Metzger approach not a numerical solution?

Page 16, line 6: take out 'only'

The text regarding "Kindly" provided emissions seems like language that should be in the acknowledgements.

---

## Author Comment (AC2) · 17 Sep 2018

**Response to Referee 1**

S. Metzger et al.

16 August 2018

We thank the referee for her/his constructive comments and manuscript review. Please find our point-by-point reply below. The MS will be revised accordingly. We hope to have satisfactorily addressed all comments.

Reply to comments.

1. *In Figure 8 the authors find that aerosol water mass concentrations are highest in the western desert of the U.S. Liao and Seinfeld 2005 and Carlton and Turpin 2013 find it is highest in the eastern U.S. where sulfate mass concentrations and RH are higher. Can the authors provide a context for this discrepancy?*

   Our water mass results are lowest in the western desert of the U.S. in agreement with Liao and Seinfeld 2005 and Carlton and Turpin 2013. Please note the inversion of the color scale. We apologize for the missing note in the caption of figure 8. We will include it in the revised MS.

   *The authors stress the importance of "aging" in their manuscript and this is not well described. I think they mean changing particle hygroscopicity with time but this is not clearly stated. Also, "aging" is a not a precise term. Do they mean increased oxidation due to longer OH exposure in the atmosphere, and the subsequent changing chemical composition important water uptake?*

   We agree, "aging" is a not a precise term. We will replace it throughout the work with "chemical aging" and add that this indeed changes the (bulk) particle hygroscopicity with time. In our model, the uptake of inorganic acids on bulk compounds, and the associated neutralization reactions and water uptake, occurs over time, i.e., during aerosol transport. This will be clarified in the revised MS.

   *The authors do not make a compelling case in the introduction for their work and I found this confusing. Below I provide comments that think help address my concerns and I hope the authors find them useful.*

We highly appreciate these constructive comments.

*The introduction is not directly linked to the premise that aerosol water is crucial. I find it difficult to understand why the introduction starts with the importance of desertification and subsequent dust emissions to properly describe AOD when their title and abstract focus on aerosol water. Do the authors mean to say that even in arid regions, AOD is not properly described in models unless water uptake is considered? That is a compelling argument and would help to connect the introductory desert discussion with aerosol water.*

Yes, indeed, even in arid regions the AOD is not properly described in models unless water uptake is considered. The reason is the interaction of air pollution with e.g., mineral dust (the issue we recently have addressed with Abdelkader et al, 2015). The missing link is the uptake of acids on mineral dust as it can alter the ability of bulk dust to take up water vapor even at a very low ambient humidity – in case of condensing hydrochloric acid, calcium chloride can be formed over time which can cause water uptake at a relative humidity as low as 28%.

*Sentence 1: "providing realistic projections of climate change is one of the most difficult tasks of climate modelers..." I would state "Providing realistic projections of climate change is difficult due to many unknowns and large uncertainties . . . " As written the sentence is awkward, the authors seem to say, conducting climate simulations is the hardest thing for climate modelers to do.*

Yes, we agree. We will change the sentence to "Providing realistic projections of climate change is difficult due to many unknowns and large uncertainties that still exists ...".

*The authors might not be aware of this paper using actual measurements of particle-phase ions and meteorology coupled with ISORROPIA-II to calculate aerosol water to better connect surface particle mass measurements to satellite AOD by Nguyen in Geophys. Res. Letts.:. I think it would help with their argument to link the importance of aerosol water with AOD and then subsequently radiative forcing calculations important for climate modeling efforts.*

Yes, we agree. We will include Nguyen et al. in the revised MS as their work helps to link the importance of aerosol water with AOD.

*Page 4, Line 13: is dry deposition based only properties of the surface? Do different chemical species all deposit at the same rate?*

This is a typo. We will change it to "Dry deposition fluxes are calculated

as the product of the surface layer concentration and the dry deposition velocity, which reflects the efficiency of the transport to- and destruction at the surface (Ganzeveld et al., 2006)".

*Page 4, Line 9/25: Why are some subroutines listed together, while others are separated out? For example CLOUD, CVTRANS, JVAL, TROPOP, H2O, ORBIT, and RAD are listed below in a similar fashion.*

The submodel listing on line 9 was left accidentally in the MS. We will clean it up in the revised MS.

*Page 4, Line 16/17: what do "...water isoprene concentration" and "methanol water deposition. . ."? Perhaps the authors mean isoprene concentration in ocean water? Do they mean dry deposition of methanol to water? I read this sentence multiple times and I am still not sure.*

The sea-air exchange submodel (AIRSEA) calculates the transfer velocity for certain soluble tracers (e.g., DMS, isoprene, methanol). This will be clarified in the revised MS.

*Page 4, Line 20: Does "Our chemical mechanism for the troposphere is similar to the one used in poz" mean the mechanism is the same as used in Pozzer et al., 2006, cited earlier? Sometimes the authors write "poz" and "Pozzer". I am not sure if they mean the same thing.*

This is a typo. "poz" should read Pozzer et al., 2012 (Atmos. Chem. Phys., 12, 961-987, www.atmos-chem-phys.net/12/961/2012/). This will be corrected accordingly and the reference added in the revised MS.

*Page 5, last sentence and continuing to the next page: "It was shown by Metzger et al. (2016b) that the $\nu_i$-approach allows to analytically solve the gas-liquid-solid partitioning and the mixed solution water uptake by eliminating the need for numerical solutions . . ." Is the Metzger approach not a numerical solution?*

Again, a typo. "solutions" should read "iterations".

*Page 16, line 6: take out "only"*

Yep, "only" will be taken out in the revised MS.

*The text regarding "Kindly" provided emissions seems like language that should be in the acknowledgements.*

Also true; parts of the sentence will be moved to the acknowledgements.

---

## Editor Decision (ED1)

```
================================================================================

Technical comments  on Metzger et al., "Aerosol water parametrization:..."

================================================================================

title: say "Importance" for what: perhaps "... importance for climate
evaluation"

abstract: you should avoid references in the abstract (it may be used
          and published independently of the paper)

references: journal titles should be abbreviated

p1., l.9: at --> on

p1, l 15: Altogether --> Overall

p1, l. reveals --> shows/demonstrates

p2, l. 2: exists --> exist

p.2., l.9: commonly --> common (or "ubiquitous")

p.2 l 28: computationally --> computational

p 3, l 15: say "potentially can significantly speed-up.."

p.3. l 22: "for this purpose we..."

p.3, l. 29: ... regional amplification by the aerosol water uptake, it is ...

p 20., l.15: you have two "https" here?

p 20., l26: double comma
```

---

## Author Response (AR2)

**Author's Response**

S. Metzger et al.

17 November 2018

Dear Editor,

Thank you very much for your time. We have addressed all comments accordingly. The additional changes are highlighted in the marked-up version included again below. We are looking forward for publication of this revised/final manuscript version.

Kind regards,

S. Metzger, et al.

[revised manuscript text omitted]